# Characteristics of Chromophoric and Fluorescent Dissolved Organic Matter in the Nordic Seas

Anna Makarewicz[1], Piotr Kowalczuk[1], Sławomir Sagan[1], Mats A. Granskog[2],

Alexey K. Pavlov[2], Agnieszka Zdun[1], Karolina Borzycka[1], Monika Zabłocka[1]

[1]Institute of Oceanology, Polish Academy of Sciences, ul. Powstańców Warszawy 55, 81–712 Sopot, Poland

[2]Norwegian Polar Institute, Fram Centre, 9296 Tromsø, Norway

Correspondence to: Anna Makarewicz (araczkowska@iopan.gda.pl)
phone: +48 58 7311804,
fax: +48 58 5512130,

Revised manuscript: 4 May, 2018

**Abstract**

Optical properties of Chromophoric (CDOM) and Fluorescent Dissolved Organic Matter (FDOM) were characterized in the Nordic Seas including the West Spitsbergen Shelf during June–July of 2013, 2014 and 2015. The CDOM absorption coefficient at 350 nm, $a_{CDOM}(350)$ showed significant interannual variation (T test, p<0.00001). In 2013, the highest average $a_{CDOM}(350)$ values ($a_{CDOM}(350) = 0.30\pm0.12$ m$^{-1}$) were observed due to the influence of cold and low–saline water from the Sørkapp Current in the southern part of West Spitsbergen Shelf. In 2014, $a_{CDOM}(350)$ values were significantly lower (T test, p<0.00001) than in 2013 (av. $a_{CDOM}(350) = 0.14\pm0.06$ m$^{-1}$), which was associated with the dominance of warm and saline Atlantic Water (AW) in the region, while in 2015 intermediate CDOM absorption (av. $a_{CDOM}(350) = 0.19\pm0.05$ m$^{-1}$) was observed. *In situ* measurements of three FDOM components revealed that fluorescence intensity of protein–like FDOM dominated in surface layer Nordic Seas. Concentrations of marine and terrestrial humic–like DOM were very low and distribution of those components were generally vertically homogenous in the upper ocean (0–100 m). Fluorescence of terrestrial and marine humic–like DOM decreased in surface waters (0–15 m) near the sea–ice edge by dilution of oceanic waters by sea–ice melt water. The vertical distribution of protein–like FDOM was characterized by a prominent sub–surface maximum that matched the subsurface chlorophyll *a* maximum and was observed across the study area. The highest protein–like FDOM fluorescence was observed in the Norwegian Sea in the core of warm AW. There was a significant relationship between the protein–like fluorescence and chlorophyll *a* fluorescence (R$^2$=0.65, p<0.0001, n=24490), which suggests that phytoplankton was the primary source of protein–like DOM in the Nordic Seas and West Spitsbergen Shelf waters. Observed variability of selected spectral indices (spectral slope coefficient, $S_{300-600}$, carbon–specific CDOM absorption coefficient at 254 and 350 nm, SUVA$_{254}$, $a^*_{CDOM}(350)$) and non–linear relationship between CDOM absorption and spectral slope coefficient also indicate a dominant marine (autochthonous) source of CDOM and FDOM in the study area. Further, our data suggest that $a_{CDOM}(350)$ cannot be used to predict dissolved organic carbon (DOC) concentrations in the study region, however the slope coefficient ($S_{300-600}$) shows some promise to be used.

## 1. Introduction

The rapid reduction of summer sea ice in the Arctic Ocean in the past decades has various repercussions on the structure and functioning of the Arctic marine system: forcing changes in physics, biogeochemistry and ecology of this complex oceanic system (Meier et al., 2014). One of the most significant consequences of observed rapid Arctic Ocean transition is an increase in the primary productivity of the Arctic Ocean (Arrigo et al., 2008), which could potentially contribute to increased production of autochthonous (marine) dissolved organic matter (DOM) in ice free and under ice waters. The sea ice is also a source of autochthonous CDOM/DOM, (e.g. Granskog et al., 2015a; Anderson and Amon, 2015; Reteletti-Brogi et al, 2018). However, DOC produced by ice algae has limited effect on overall organic carbon mass balance in the Arctic Ocean, as melting of one meter of sea ice would negligibly change DOC concentration in top 50 m of water column, assuming an averaged DOC content in the ice of 100 µMol C (Anderson and Amon, 2015). Simultaneously, response of terrestrial ecosystems to temperature increase will accelerate permafrost thaw and increase the riverine discharge, resulting in more allochthonous (terrestrial) DOM being released into the Arctic Ocean (Amon, 2004; Stedmon et al., 2011; Anderson and Amon, 2015; Prowse et al., 2015, and references therein). Terrestrial DOM presents a considerable role in the carbon budget of the Arctic Ocean (Findlay et al., 2015; Stein and Macdonald, 2004), especially in coastal waters and continental shelf with large inflow of terrestrial DOM, which constitutes 80% of total organic carbon delivered by Arctic rivers (Stedmon et al., 2011).

The optically active DOM fraction called chromophoric or colored dissolved organic matter (CDOM) represents light absorbing molecules (Coble, 2007; Nelson and Siegel, 2013; Stedmon and Nelson, 2015). Once entered or produced in surface waters of the Arctic Ocean, CDOM has a significant influence on heating of the uppermost ocean layer and its stratification (Pegau, 2002; Hill, 2008; Granskog et al., 2007, 2015b). Particularly in absence of sea ice, light absorbed by CDOM in visible part of the spectrum limits the light available for photosynthetic organisms (Arrigo and Brown, 1996), but also shields marine ecosystem from potentially harmful ultraviolet radiation strongly absorbing electromagnetic radiation in UVB and UVA bands (Erickson III et al., 2015). CDOM is also important substrate in photochemical reactions contributing to direct remineralization of organic carbon, production of bioavailable low molecular weight DOM but also formation of reactive oxygen species that could potentially be toxic to marine organisms (Mopper and Kieber, 2002, Kieber et al., 2003,

Zepp, 2003). The mineralization by photochemical reactions or microbes of DOM both terrestrial and marine is a crucial but still insufficiently quantified mechanism in the Arctic carbon cycle (e.g. Osburn et al., 2009). Despite the importance of CDOM, studies on its distribution, properties and transformation in the Arctic Ocean and its marginal seas are still limited, partly by their remoteness and seasonal accessibility.

A sub–fraction of CDOM fluoresces and is called fluorescent dissolved organic matter (FDOM). Recent advances in fluorescence spectroscopy (Coble, 1996) and data analyses techniques has provided a more comprehensive overview of FDOM characteristics. Based on excitation/emission spectra fluorescence spectroscopy it is possible to distinguish amidst different origin groups of fluorophores e.g. terrestrial, marine and anthropogenic (Stedmon et al., 2003, Murphy et al., 2013; Murphy et al., 2014). Use of *in situ* DOM fluorometers enables low cost and high sample rate observations of distribution of FDOM and related biogeochemical proxies with greater temporal and spatial resolution (Belzile et al., 2006; Kowalczuk et al., 2010).

North Atlantic sector of the Arctic Ocean is a region with complex interaction of inflowing warm and highly productive Atlantic Waters entering the Arctic and cold and fresh Polar Surface Waters exiting the Arctic Ocean. Recent studies have reported intensification of Atlantic Water (AW) inflow into Arctic Ocean (Walczowski 2014; Polyakov et al., 2017; Walczowski et al., 2017) further highlighting the importance of the European sector of the Arctic Ocean to better understand the complex interactions between inflowing AW and Polar Waters. Optically these waters are contrasting, especially with respect to CDOM (Granskog et al., 2012; Pavlov et al., 2015, Stedmon et al., 2015) and FDOM (Jørgensen et al., 2014; Gonçalves-Araujo et al., 2016). In absence of sea ice, favorable vertical mixing conditions and sufficient levels of solar radiation makes it a very productive and important region from an ecosystem and socio–economic standpoint, thus ensuring motivation for ongoing studies of the complex marine system in the area (Skogen et al., 2007; Olsen et al., 2009; Dalpadado et al., 2014). In context of ongoing and further anticipated intensification of Atlantic Ocean inflow to the Arctic Ocean, description of processes and factors controlling CDOM/FDOM properties and distribution could be used to better predict future changes associated with CDOM in the areas upstream of the Atlantic Water inflow region, to estimate of glacial melt water (Stedmon et. al.,2015) and to trace water masses (Gonçalves-Araujo et al., 2016).

A number of occasional synoptic surveys of CDOM and optical properties have been conducted in the different regions of the European Arctic Ocean and concentrated on the

western part of the Fram Strait influenced by Polar Water outflow with EGC (Granskog et al., 2012; Pavlov et al., 2015; Gonçalves–Araujo et al., 2016). The CDOM distribution in the area influenced by AW were reported by Stedmon and Markager (2001) in the central part of the Greenland Sea, and by Granskog et al. (2012) and Pavlov et al. (2015) who presented CDOM and particulate absorption distribution along transect across Fram Strait at 79°N. Hancke et al. (2014) studied seasonal distribution of CDOM absorption coefficient ($a_{CDOM}(\lambda)$) in an area across the Polar Front in the central part of the Barents Sea. Seasonal studies on CDOM contribution to overall variability of inherent optical properties (IOPs) were reported in sea ice (Kowalczuk et al., 2017) and in the water column during a spring under–ice phytoplankton bloom north of Svalbard (Pavlov et al., 2017). In this study we aimed to present variability of CDOM and FDOM optical properties in a large area spanning parts of the Barents, Norwegian and Greenland Seas (particularly focusing on West Spitsbergen Shelf) over a period of three consecutive years (2013–2015) and understand the role of *i*) large scale ocean circulation patterns and water mass distribution and *ii*) phytoplankton productivity as controlling factors on CDOM and FDOM distribution.

## 2. Material and Methods

### 2.1. Study area

Observations were done in the framework of long term observational program AREX, conducted since 1987 by Institute of Oceanology Polish Academy of Science, Sopot, Poland and covered the area of water masses exchange between the North Atlantic Ocean and the Arctic Ocean (Figure 1). The Norwegian, Barents and Greenland Seas, called Nordic Seas, represents a crucial component of the Northern Hemisphere climate system due to two contrasting water masses and their contribution to the heat and salt exchanges between the North Atlantic and the Arctic Ocean (Walczowski, 2014; Schlichtholz and Houssais, 1999a, b). The warm and salty Atlantic Waters (AW) carried northward by the North Atlantic Current (NAC), which further splits into two major branches. Norwegian Current (NC) flows into the Barents Sea as the Barents Sea Branch, while the West Spitsbergen Current (WSC) heads north along the eastern flank of the Fram Strait. The East Greenland Current (EGC) flows south along the western side of Fram Strait, and carries cold and low saline Polar Surface Waters (PSW) and sea ice (Figure 1) (e.g. Schlichtholz and Houssais, 2002). East Spitsbergen Current (ESC) could also affect the region with transformed Polar Water originating from north–east Barents Sea (Sternal at al., 2014). Main ESC branch flows southward along the coast of Spitsbergen and its extension is the Sørkapp Current that

influences the West Spitsbergen Shelf. Remaining part of Polar Water from the Barents Sea flows southwestward along the eastern slope of the Spitsbergenbanken (SPB) towards Bear Island as the Bjørnøya Current (BC) (Loeng, 1991) in the Norwegian Sea and the Barents Sea border. Presence and extensiveness of Polar Water from the Barents Sea depends on favorable wind conditions affecting the magnitude and the exchange with the AW inflow (Nilsen et al., 2015; Walczowski, 2014).

Optical measurements and water sampling were conducted during three summer Arctic expeditions (AREX) onboard r/v Oceania in 2013, 2014 and 2015 (AREX2013, AREX2014, and AREX2015, respectively) (Table 1). *In situ* FDOM fluorescence measurements were conducted in 2014 and 2015. AREX expeditions covered the Norwegian Sea with a main section along the border between the Norwegian Sea and the Barents Sea (sampled in late June to early July in 2014 and 2015). The area of the western and northern Spitsbergen shelf was investigated in July of 2013–2015 (Figure 1), along sections spanning from shelf towards the sea–ice edge. The westernmost and northernmost sampling stations north of 76°N, shown on Figure 1, corresponds to the sea ice edge position in July in the given year.

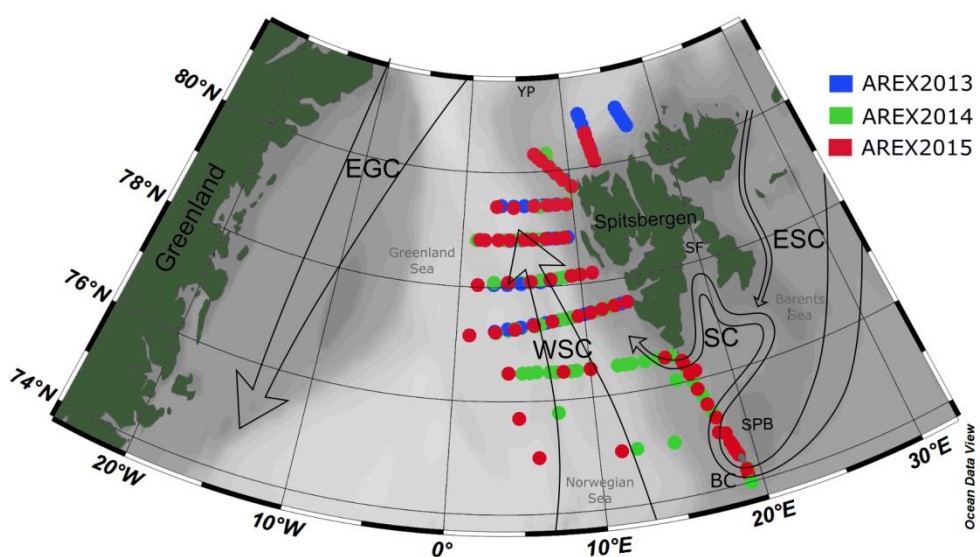

Figure 1: Map of the sampling stations during AREX2013 (blue circles), AREX2014 (green circles), AREX2015 (red circles) with general surface circulation patterns in the Nordic Seas. Atlantic Waters: WSC, West Spitsbergen

 Current. Polar waters: ESC, East Spitsbergen Current; SC, Sørkapp Current;

EGC, East Greenland Current; BC, Bjørnøya Current; YP, Yermak Plateau;
SF, Storfjorden; SPB, Spitsbergenbanken.
**Table 1.** Dates of AREX expeditions, and number of samples or number of *in situ*
vertical profiles of CDOM, DOC, chlorophyll *a, Chla,* inherent optical
properties (IOP), chlorophyll a fluorescence ($I_{FChla}$), and FDOM fluorescence.

| Cruise | Date | Water samples | | Instrumental measurements | |
|---|---|---|---|---|---|
| | | CDOM / DOC | *Chla* | IOP and $I_{FChla}$ | FDOM |
| | | N samples | | N profiles | N profiles |
| AREX2013 | 13.07–24.07.2013 | 79 | 78 | 57 | 0 |
| AREX2014 | 20.06–23.07.2014 | 221 | 138 | 100 | 100 |
| AREX2015 | 19.06–24.07.2015 | 263 | 142 | 68 | 68 |

*2.2. Sample collection and processing*
Water samples for determination of CDOM absorption, chlorophyll *a* and DOC were
collected with SeaBird SBE32 Carousel Water Sampler equipped with Niskin bottles, the
SBE 911 *plus* CTD probe (SBE 9plus CTD Unit and SBE 11plus Deck Unit) and Wetlabs
ECO Chlorophyll fluorometer. Samples were collected at three depths: near the surface, ca. 2
m depth, at chlorophyll *a* maximum, that was usually located between 15 and 25 m depth, and
below chlorophyll *a* maximum, between 50 and 70 m. The exact position of chlorophyll *a*
maximum depth was estimated from vertical profile of chlorophyll *a* fluorescence during
CTD downcast. During AREX2013 water samples for CDOM absorption measurements were
immediately filtered in two steps: firstly through acid–washed GF/F filters, and secondly
through acid–washed Sartorius 0.2 μm pore size cellulose membrane filters to remove finer
particles. In 2014 and 2015 CDOM samples were filtered directly from rosette Niskin bottles
through a Millipore Opticap XL4 Durapore filter cartridge with nominal pore size 0.2 μm into
acid washed 200 ml amber glass bottles. The cartridge filter was kept in 10% HCl solution
and was rinsed with ultrapure MilliQ and sample water before collecting CDOM samples. In
2013 and 2015 collected unpreserved water samples for determination of CDOM absorption
were stored on board r/v Oceania in dark, at temperature of 4°C, and were transferred after the
cruise to land–based laboratory for spectroscopic measurements. In 2014, all spectroscopic
measurements for determination of CDOM absorption were done in the laboratory onboard
r/v Oceania, immediately after collection. Samples for determination of DOC concentration
were collected the same way as CDOM samples. Water that passed through 0.2 μm filters was
collected into pre–cleaned 40 ml glass vials (certified pre–cleaned sample vials, Sigma–
Aldrich) and acidified with drop of concentrated 38% HCl. Acidified samples were stored on
board ship in dark, at temperature of 4°C, and were transferred after the cruise to land–based
laboratory for measurements.

Water samples for determination of chlorophyll *a* concentration were filtered

immediately after collection under low vacuum on Whatman (GE Healthcare, Little Chalfont,
UK) 25 mm GF/F filters. Filter pads with particulate material retained on them, were
immediately deep frozen in a freezer and thereafter stored at -80°C prior to analyses.
*2.3. CDOM absorption*

Before spectroscopic scans were conducted, the temperature of the CDOM absorption

samples was increased to room temperature. CDOM absorption of AREX2013 and
AREX2015 was measured using a double–beam Perkin Elmer Lambda 650
spectrophotometer in the spectral range 240–700 nm, in the laboratory at the Institute of
Oceanology Polish Academy of Sciences in Sopot, Poland. Measurements of the CDOM
absorption samples collected during AREX2014 were done on board of research vessel, using
a double–beam Perkins Elmer Lambda 35 spectrophotometer in the same spectral range as in
2013 and 2015. The 10–cm quartz cuvette was chosen for all measurements and the reference
was fresh ultrapure water. Absorbance $A(\lambda)$ spectra were transformed to the CDOM
absorption coefficients, $a_{CDOM}(\lambda)$ [m$^{-1}$], according to:

$$a_{CDOM}(\lambda) = 2.303 \cdot A(\lambda)/L \qquad (1)$$

where, 2.303 is the natural logarithm of 10, $A(\lambda)$ is the corrected spectrophotometer
absorbance reading at a specific wavelength $(\lambda)$ and $L$ is the path length of optical cell in
meters (here 0.1 m).

Slope coefficient of the CDOM absorption spectrum, $S$, between 300 and 600 nm was

derived using Equation (2) and was implemented in Matlab R2011b by adopting a nonlinear
least squares fit with a Trust–Region algorithm (Stedmon et al., 2000, Kowalczuk et al.,

2006):

$$a_{CDOM}(\lambda) = a_{CDOM}(\lambda_0)e^{-S(\lambda_0 - \lambda)} + K \qquad (2)$$

where: $\lambda_0$ is a reference wavelength (here 350 nm), and $K$ is a background constant
representing any possible baseline shifts not due to CDOM absorption. Simultaneous
calculation of three parameters: $a_{CDOM}(350)$, $S$, and $K$ were done according to Equation (2) in
the spectral range between 300 and 600 nm by non–linear regression. CDOM absorption
coefficient values are also included at two other wavelengths: $a_{CDOM}(375)$ and $a_{CDOM}(443)$ to
enable direct comparison of our results with previously published studies. In 2014 the range
of spectral slope coefficient had to be reduced to 300–500 nm due to spectra disturbances
over 500 nm in data set from the western and northern Spitsbergen shelf. To assess the effect
of the narrower spectral range on spectral slope coefficient calculations we calculated slopes
for both spectral ranges in 2013 and 2015. On average, spectral slope coefficient in the
spectral range 300–500 nm was higher by 1.76 $\mu m^{-1}$ relative to $S_{300-600}$. Calculated average
bias was deduced from $S_{300-500}$ calculated in 2014 to comply with 2013 and 2015 data set.
Linear regression model was used on log–transformed CDOM absorption spectra for spectral
slope coefficient calculations at spectral range 275–295 nm, $S_{275-295}$.
*2.4. Chlorophyll a concentration.*
Filters pads containing suspended particles (including pigments) were used for
determination of the chlorophyll *a* concentration for all AREX cruises. Pigments were
extracted at room temperature in 96% ethanol for 24 hours. Spectrophotometric determination
of chlorophyll *a* concentration, *Chla* [mg m$^{-3}$], was done with two spectrophotometers: UV4–
100 (Unicam, Ltd) and with a Perkin Elmer Lambda 650 in 2013 and 2014–2015,
respectively. The optical density (absorbance) of pigment extract in ethanol was measured at
665 nm. Background signal was corrected in the near infrared (750 nm):
$\Delta OD = OD(665nm) - OD(750nm)$. Subsequently, conversion of absorbance to chlorophyll *a*
was done according to following equation (Strickland and Parsons, 1972; Stramska et al.,

2003):

$$Chla = (10^3 \cdot \Delta OD \cdot V_{EtOH})/(83 \cdot V_w \cdot l). \qquad (3)$$
where: 83 [dm$^3$ (g cm)$^{-1}$], is chlorophyll *a* specific absorption coefficient in 96% ethanol, $V_w$
[dm$^3$] is the volume of filtered water , $V_{EtOH}$ [dm$^3$] is ethanol extract volume, and the *l* is path
length of cuvette (here 2 cm ).
*2.5. DOC concentration*
DOC measurements were done with a 'HyPer+TOC' analyzer (Thermo Electron
Corp., The Netherlands) using UV persulphate oxidation and non–dispersive infrared
detection (Sharp, 2002). Potassium hydrogen phthalate was used as standard addition
measurements method for each sample in triplicate. Consensus reference material (CRM)
supplied by Hansell Laboratory from University of Miami was analyzed as a quality control
of DOC concentrations. The methodology provided sufficient accuracy (average recovery
95%; n = 5; CRM = 44 − 46 μM C; our results = 42 − 43 μM C) and precision represented by
a relative standard deviation (RSD) of 2%.
The carbon–specific CDOM absorption coefficient at 350 nm, $a^*_{CDOM}(350)$ [$m^2g^{-1}$],
was determined as the ratio of the CDOM absorption coefficient at a given wavelength
$a_{CDOM}(350)$ to the DOC concentration (Equation 4):
$$a^*_{CDOM}(350) = \frac{a_{CDOM}(350)}{DOC}$$
(4),

where DOC is expressed in mg $l^{-1}$.
The carbon–specific UV absorption coefficient (SUVA) is defined as the UV
absorbance of water sample at specific wavelength normalized for DOC [mg $l^{-1}$]
concentration (Weishaar et al., 2003). SUVA [$m^2$ $gC^{-1}$] at 254 nm (SUVA$_{254}$, Equation 5) is
an indicator of aromaticity of aquatic humic substances and was calculated as:
$$SUVA_{254} = \frac{a_{CDOM}(254)}{DOC}$$
(5).

*2.6. Instrumental in situ measurements of inherent optical properties, FDOM and*

*chlorophyll a fluorescence*
Vertical profiles of inherent optical properties (IOP), FDOM and chlorophyll *a*
fluorescence together with conductivity, temperature and pressure were measured at all
stations from the surface down to 200 m depth using an integrated instrument package
consisting of an ac–9 *plus* attenuation and absorption meter (WET Labs Inc., USA), a
WetStar CDOM fluorometer (WET Labs Inc., USA), a MicroFlu–Chl chlorophyll *a*
fluorometer (TrioS GmbH, Germany), and a Seabird SBE 49 FastCAT Conductivity–
Temperature–Depth probe (Seabird Electronics, USA).
Spectral light absorption, $a(\lambda)$ and beam attenuation, $c(\lambda)$, coefficients were measured
at nine wavelengths (412, 440, 488, 510, 532, 555, 650, 676, and 715 nm). The ac–9 *plus*
calibrations were performed regularly. After cleaning with ultrapure water, stability
instruments readings were inspected with in-air measurements. The required correction of
absorption signal for scattering was performed with so–called proportional method where
zero absorption is estimated at 715 nm (Zaneveld et al., 1994). Subtraction of absorption
coefficients from attenuation coefficients determined volume scattering coefficient, $b(\lambda)$.
Excitation channel and the maximum emission of light detector of the MicroFlu–Chl
chlorophyll *a* fluorometer were set at 470 nm and at 686 nm, respectively. Recorded
chlorophyll *a* fluorescence intensity signals, $I_{FChla}$ were reported as analog voltage output in
the range 0–5 V DC. The instrument setup is described in detail in Granskog et al. (2015b).
FDOM was measured using a 3–channel WETLabs WetStar fluorometer equipped
with two laser LEDs that excited the water sample inside the flow–through quartz cell at 280
and 310 nm, and two detectors to measure emission intensity at 350 and 450 nm. Such
construction allowed for combinations of three channels with distinct excitation/emission
features in specific peak areas as given in Coble (1996): Channel 1 (CH1), ex./em. 310/450
nm, represents marine ultraviolet humic–like peak C and marine humic–like peak M; Channel
2 (CH2), ex./em. 280/450 nm, represents UVC terrestrial humic–like peak A; and Channel 3
(CH3), ex./em. 280/350 nm, represents the protein–like tryptophane peak T (Figure S1). $I_{CHn}$
is the fluorescence intensity at particular channel where *n* denotes the channel number from 1
to 3. Recorded $I_{CHn}$ could be transformed from raw instrument counts into either the quinine
sulfate equivalent (QSE) units, or particular compounds concentration with factory calibration
curves. Application of the factory calibration curves, especially the blank ultrapure water
readings offset resulted in negative values for $I_{CH1}$ and $I_{CH2}$. Therefore, we reported
fluorescence intensities acquired from the WetStar fluorometer in raw counts, (R.C.)
corrected for a noticeable but small drift. This offset was determined as the difference in any
$I_{CHn}$, between initial measurements in July 2014 in the depth range 100–150 m, at salinity
>34.9 and temperature T >0°C and measurements repeated in the same salinity and
temperature range during field campaign in 2015. The water salinity and temperature
characteristics at the chosen depth range was typical for core of Atlantic water inflow, which
is characterized with stable values of spectral absorption (measured with ac–9 *plus* attenuation
and absorption meter), negligible chlorophyll *a*, and very low background CDOM absorption
level (Sagan S., *personal communication,* 2017). Therefore, we assume that any differences in
raw WETLabs WetStar 3–channel fluorometer readings between measurements in 2014 and
2015 resulted from instrument drift, and the offset between the years has been subtracted from
florescence intensity values at each channel measured in 2015.
*2.7. Classification of water masses*
Water masses were classified according to Rudels at al. (2005) based on potential
temperature (Θ), potential density ($\sigma_\theta$) and salinity (S). The original classification definitions
are derived for Fram Strait (Rudels et al., 1999) and categorization used in Rudels et al.
(2002, 2005) considers mainly the EGC, the area of Yermak Plateau and Storfjorden located
on the east coast of Spitsbergen. To adjust the classification to the broader area of Nordic
Seas including Atlantic part (Norwegian and Barents Sea) some modifications have been
introduced (see Table S1).
The epipelagic layer of the Nordic Seas is dominated by AW and PSW, and waters
formed in the mixing process and local modifications (precipitation, sea–ice melt, riverine
run–off, and surface heating or cooling) of these two water masses. AW masses were usually
characterized by potential temperature and density thresholds defined by Rudels et al. (2005)
(Table S1). To better distinguish AW from PSW, we added a third criterion: any water mass
classified as PSW (Rudels et al., 2005) with salinity higher than S>34.9, has been considered
as AW. The salinity criterion equal to 34.9 is widely used in the literature (Swift and Aagaard
1981; Schlichtholz and Houssais 2002; Walczowski 2014) and eliminates Rudels' et al.
(2005) classification ambiguity caused by modification of AW by local sources of fresh
water. Part of AW (except Polar Surface Water warm, PSWw) included waters with density
below $\sigma_\theta=27.7$ kgm$^{-3}$ (marked on Figure 3 with dashed isopycnal line) used by Rudels et al
(2005) as a threshold value between AW and PW. Lower density of waters of Atlantic domain
with high salinity (>34.9) is predominantly caused by high temperatures and cannot be
referred to as PSW, which lower density is attributed to lower salinity. Polar Surface Water
(PSW) is defined as $\Theta \leq 0°C$ and $\sigma_\theta \leq 27.7$ kgm$^{-3}$. The temperature of PSW is usually negative,
however, positive temperatures (3–5°C) can be observed during summer (Swift and Aagaard
1981). Warmer PSWw has been considered here with the same $\sigma_\theta \leq 27.7$ kg m$^{-3}$ criterion and
$\Theta > 0°C$ (Rudels et al., 2005), due to summer season measurements and higher temperatures of
low salinity surface waters in the eastern Fram Strait. Furthermore PSWw was also limited to
the uppermost 50 m of the water column with $S \leq 34.9$. The water mass with similar TS
characteristics to PSWw but slightly different ranges were referred to in the literature for
Arctic Surface Water, ASW (e.g. Pavlov et al., 2015 and Gonçalves-Araujo et al., 2016), but
due to the dominance in the area of water originating from Atlantic Ocean the name PSWw
from Rudels et al. (2005) classification is used. We could find Arctic Atlantic Water (AAW)
in our data set as a result of mixing process of AW and PW, in the range of $0 < \Theta \leq 2°C$ and
$27.7 < \sigma_\theta \leq 27.97$ (Rudels et al., 2005). Arctic Intermediate Waters (AIW) was defined as
$\Theta \leq 0.3°C$, $27.97 < \sigma_\theta$, $\sigma_{0.5} \leq 30.44$ (Rudels et al., 2005) and included measurements taken at
greatest depth in this study.

## 3. Results

*3.1. Interannual and spatial variability of CDOM properties in surface waters with relation to hydrography.*

Spatial distribution of temperature, salinity and $a_{CDOM}(350)$ in surface waters of West Spitsbergen Shelf and Norwegian Sea shows considerable variation between years (Figure 2). In 2013, the West Spitsbergen Shelf was under the influence of cold and low saline waters from SC. The impact of this current together with possible terrestrial runoff (highest $a_{CDOM}(350)$ values were observed at Spitsbergen fjords entrances) was reflected in high $a_{CDOM}(350)$ (av. $0.47\pm0.26$ m$^{-1}$) for coastal waters on the West Spitsbergen Shelf. Lower values of $a_{CDOM}(350)$ were observed in the PSWw (av. $0.33\pm0.17$ m$^{-1}$) from coastal areas and in the warm and salty AW from the WSC (av. $0.28\pm0.07$ m$^{-1}$). The lowest CDOM absorption (av. $0.25\pm0.05$ m$^{-1}$) in 2013 was observed at the northernmost and northeastmost stations influenced by low saline PSW affected by sea–ice melt water.

A quite different situation was observed in 2014 (Figure 2b). The spatial extent of AW was distinctly wider, as shown by temperature and salinity distributions. The higher proportion of AW over the West Spitsbergen Shelf in 2014 was confirmed by the temperature and salinity time series in the top 200 m water layer (Walczowski et al., 2017). This large volume of AW influenced CDOM absorption, which was lowered to half of the values (av. $a_{CDOM}(350) = 0.15\pm0.06$ m$^{-1}$) compared to 2013. Besides, mean $a_{CDOM}(350)$ values around $0.1\pm0.03$ m$^{-1}$ were observed in the northern Spitsbergen shelf in the area affected by sea ice melting (within the salinity range of 31.4–33.9).

In 2015, SC and ESC branches originating from the Barents Sea were pronounced, as indicated by lower temperature and salinity, Figure 2c, resulting in elevated $a_{CDOM}(350)$ values on the West Spitsbergen Shelf and along the section from Sørkapp down to 74°N and near Bjørnøya Island. In 2015 AW was characterized by low CDOM concentration ($a_{CDOM}(350)$ av. $0.17\pm0.02$ m$^{-1}$) in contrast to PSW observed north of Svalbard (av. $a_{CDOM}(350) = 0.27\pm0.05$ m$^{-1}$).

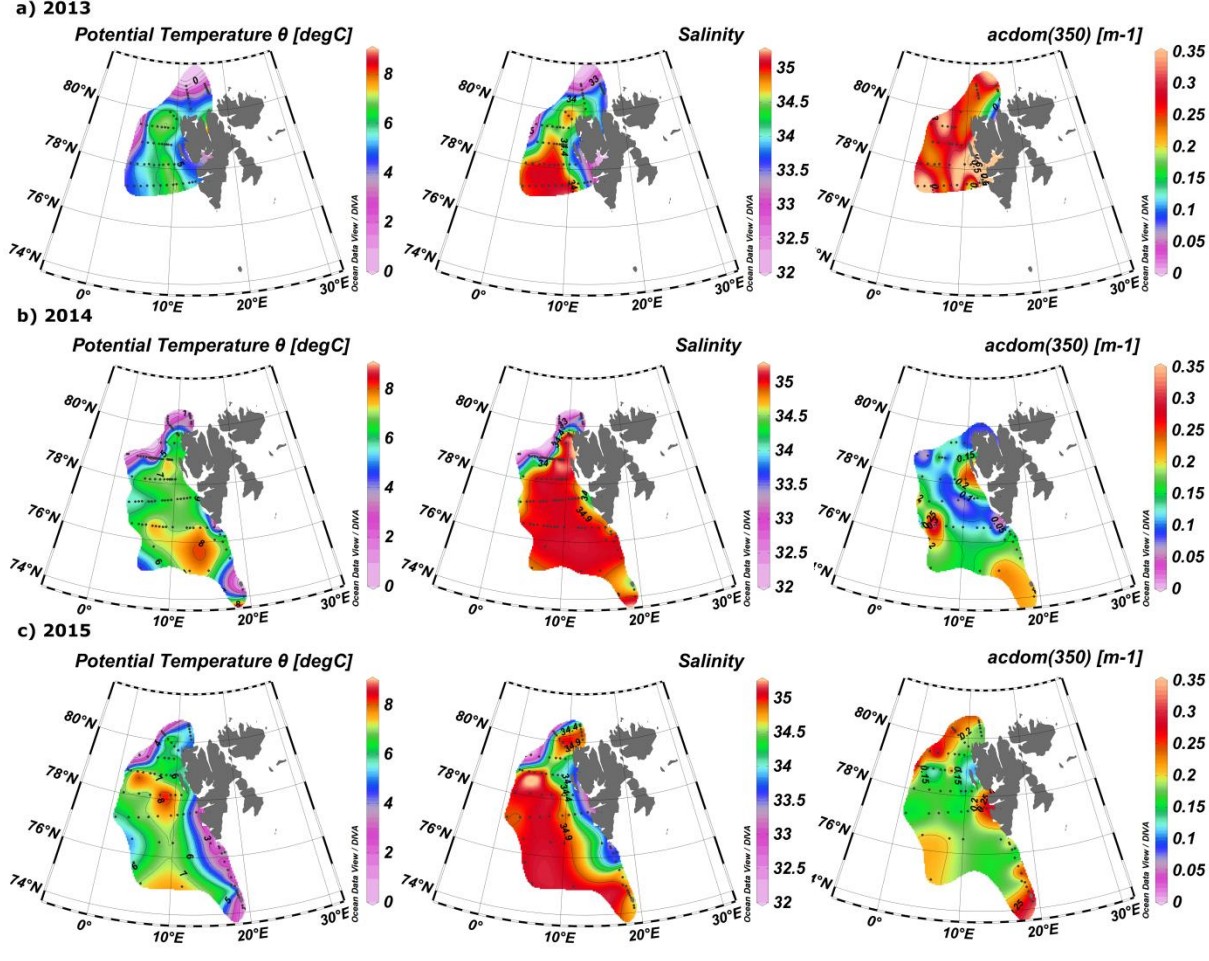

Figure 2: Surface distribution of temperature, salinity and $a_{CDOM}(350)$ in 2013–2015 (A–C respectively). Plots were created with use of Ocean Data View (Schlitzer, R., Ocean Data View, http://odv.awi.de, 2016.)

Summary statistics of the variability of $a_{CDOM}(350)$, $a_{CDOM}(443)$, $S_{275–295}$, $S_{300–600}$, $a*_{CDOM}(350)$ and $SUVA_{254}$ in different water masses in a given year is provided in Table 2. The highest $a_{CDOM}(350)$ was observed in 2013 (Table 2) when CDOM absorption in AW and PSW were similar (av. $a_{CDOM}(350) = 0.28\pm0.07$ m$^{-1}$). CDOM absorption in PSWw was higher and was characterized by the greatest variability (av. $a_{CDOM}(350) = 0.32\pm0.16$ m$^{-1}$; min–max: 0.15–0.9 m$^{-1}$ C.V. = 50%). In 2014 $a_{CDOM}(350)$ values were almost 2 times lower compared to other summer seasons (Table 2). In 2014 79% of all samples were classified as AW (av. $a_{CDOM}(350) = 0.14\pm005$ m$^{-1}$) which corresponded to the highest temperature, widespread AW distribution and lack of apparent influence by SC waters. Less

than 15% samples represented PSWw (av. $a_{CDOM}(350) = 0.14\pm0.05$ m$^{-1}$) (Table 2). In 2015 we observed intermediate $a_{CDOM}(350)$ values in AW and PSWw (Table 2) with the highest values in PSW and AAW (PSW: $a_{CDOM}(350) = 0.26\pm0.09$ m$^{-1}$, AAW: $a_{CDOM}(350) = 0.25\pm0.06$ m$^{-1}$).

The spectral slope coefficient is often inversely non–linearly related to CDOM absorption coefficient (Stedmon and Markager 2001; Stedmon et al., 2003, Kowalczuk et al., 2006, Meler et al., 2016). $S_{275-295}$ and $S_{300-600}$ was lowest in 2013 and highest in 2014, with intermediate values in 2015 (Table 2). The carbon–specific CDOM absorption coefficient $a^*_{CDOM}(350)$ was significantly lower ($p<0.000001$, T test) in 2014 compared to 2013 and 2015. While the values of SUVA$_{254}$ were most diverse in 2013 whereas the greatest variability in AW (min–max: 0.64–9.23 m$^2$ gC$^{-1}$) was observed in 2014. In 2014 and 2013 average values of SUVA$_{254}$ for whole season were similar, around 1.7 m$^2$ gC$^{-1}$ (Table 3), however average values in AW and PSWw were higher in 2013 and 2014, respectively (Table 2). In 2015 average SUVA$_{254}$ values were similar within identified water masses and low variation ($\pm$ 0.15 m$^2$ gC$^{-1}$) between different waters was observed. The interannual variability of SUVA$_{254}$ was insignificant ($p=0.89$, T test) between 2013 and 2014, however the average SUVA$_{254}$ values observed in 2015 were significantly different ($p<0.002$, T test) than in 2013 and 2014 (Table 2).

The average DOC concentration in the study area was highest in 2013 (80.69 μmol/L) and decreased significantly ($p<0.000001$, T test) year by year (Table 3) to 67.64 μmol/L in 2015. The average chlorophyll $a$ concentration was lowest in 2013 (0.87 mg/m$^3$), almost doubled in 2014 (1.58 mg/m$^3$), and decreased by 12% in 2015 (1.39 mg/m$^3$), relative to previous year.

Table 2. Descriptive statistics of selected parameters from AREX 2013–2015. Average and standard deviation, range of variability in
depth, potential temperature (Θ), salinity (S), absorption coefficient at 350 nm ($a_{CDOM}(350)$), absorption coefficient at 443 nm
($a_{CDOM}(443)$), spectral slope coefficient in range 275–295 nm ($S_{275-295}$), spectral slope coefficient in range 300–600 nm ($S_{300-600}$).
Water masses were classified according to Rudels et al. (2005) with minor modifications (see Table S1).

| WM/N | Depth [m] | Θ [°C] | S | $\sigma_\theta$ [kg*m⁻³] | $a_{CDOM}(350)$ [m⁻¹] | $a_{CDOM}(443)$ [m⁻¹] | $S_{275-295}$ [µm⁻¹] | $S_{300-600}$ [µm⁻¹] | $a^*_{CDOM}(350)$ [m² g⁻¹] | SUVA₂₅₄ [m² gC⁻¹] |
|---|---|---|---|---|---|---|---|---|---|---|
| AREX 2013 | | | | | | | | | | |
| AW n=43 | **31 ±23** | **4.94 ±1.3** | **35.01 ±0.06** | **27.68 ±0.15** | **0.28 ±0.07** | **0.05 ±0.02** | **15.36 ±3.40** | **18.25 ±1.78** | **0.35 ±0.12** | **1.95 ±0.60** |
| | 0 80 | 2.15 7.48 | 34.82 35.10 | 27.34 27.95 | 0.19 0.55 | 0.03 0.14 | 10.53 25.38 | 13.64 20.79 | 0.15 0.60 | 1.01 3.16 |
| PSW n=3 | **23 ±25** | **-0.86 ±0.7** | **33.62 ±1.00** | **27.04 ±0.84** | **0.28 ±0.03** | **0.05 ±0.00** | **16.02 ±2.35** | **17.69 ±2.15** | **0.24 ±0.02** | **1.31 ±0.28** |
| | 0 50 | -1.35 -0.02 | 32.50 34.42 | 26.09 27.70 | 0.24 0.30 | 0.05 0.06 | 14.26 18.69 | 15.21 19.07 | 0.22 0.25 | 1.00 1.55 |
| PSWw n=33 | **4 ±9** | **4.87 ±1.6** | **34.21 ±0.66** | **27.05 ±0.45** | **0.32 ±0.16** | **0.07 ±0.07** | **15.37 ±3.16** | **17.55 ±3.58** | **0.29 ±0.11** | **1.64 ±0.72** |
| | 0 30 | 0.15 7.30 | 32.21 34.89 | 25.83 27.66 | 0.15 0.90 | 0.01 0.32 | 11.61 28.32 | 9.95 30.06 | 0.15 0.58 | 0.95 3.80 |
| AREX 2014 | | | | | | | | | | |
| AW n=174 | **39 ±39** | **5.57 ±1.2** | **35.03 ±0.05** | **27.62 ±0.14** | **0.14 ±0.06** | **0.02 ±0.02** | **14.66 ±2.19** | **20.98 ±5.42** | **0.16 ±0.08** | **1.79 ±1.33** |
| | 0 200 | 2.05 7.45 | 34.86 35.09 | 27.36 27.94 | 0.04 0.34 | 0.00 0.09 | 11.20 24.52 | 10.83 42.26 | 0.05 0.59 | 0.64 9.23 |
| PSW n=4 | **15 ±12** | **-0.62 ±0.4** | **32.59 ±1.33** | **26.19 ±1.09** | **0.11 ±0.04** | **0.01 ±0.01** | **12.20 ±0.40** | **22.08 ±4.91** | **0.15 ±0.05** | **1.96 ±0.63** |
| | 5 25 | -0.91 -0.01 | 31.29 33.88 | 25.14 27.25 | 0.08 0.16 | 0.01 0.02 | 11.80 12.71 | 17.03 28.35 | 0.09 0.20 | 1.26 2.76 |
| PSWw n=28 | **18 ±15** | **2.82 ±1.9** | **34.14 ±0.73** | **27.19 ±0.54** | **0.14 ±0.05** | **0.02 ±0.01** | **13.89 ±2.42** | **20.03 ±4.72** | **0.17 ±0.07** | **1.62 ±0.78** |
| | 5 50 | 0.34 5.83 | 32.41 34.88 | 25.94 27.70 | 0.05 0.29 | 0.00 0.07 | 10.51 21.40 | 13.18 33.79 | 0.05 0.38 | 0.76 3.81 |
| AAW n=4 | **80 ±24** | **1.36 ±0.5** | **34.86 ±0.05** | **27.91 ±0.05** | **0.15 ±0.05** | **0.02 ±0.01** | **16.56 ±5.58** | **20.32 ±0.46** | **0.15 ±0.08** | **1.44 ±0.81** |
| | 50 100 | 0.59 1.89 | 34.83 34.94 | 27.86 27.97 | 0.10 0.20 | 0.01 0.02 | 12.45 24.28 | 19.77 20.87 | 0.08 0.26 | 0.67 2.31 |
| IW/DW n=11 | **1627 ±979** | **-0.66 ±0.3** | **34.94 ±0.04** | **28.09 ±0.02** | **0.17 ±0.08** | **0.03 ±0.03** | **16.46 ±5.85** | **17.83 ±4.58** | **0.17 ±0.09** | **1.07 ±0.26** |
| | 301 2823 | -0.86 -0.07 | 34.91 35.01 | 28.08 28.15 | 0.06 0.32 | 0.00 0.10 | 10.66 26.04 | 11.13 28.35 | 0.05 0.37 | 0.56 1.38 |
| AREX 2015 | | | | | | | | | | |
| AW n=156 | **61 ±65** | **4.89 ±1.5** | **35.00 ±0.06** | **27.68 ±0.15** | **0.18 ±0.04** | **0.03 ±0.01** | **19.42 ±2.55** | **19.77 ±2.15** | **0.21 ±0.05** | **1.41 ±0.24** |
| | 5 470 | 2.23 8.15 | 34.78 35.09 | 27.26 27.97 | 0.11 0.34 | 0.01 0.10 | 10.94 25.51 | 13.08 25.48 | 0.14 0.39 | 0.86 2.19 |
| PSW n=6 | **32 ±11** | **-0.58 ±0.6** | **34.14 ±0.22** | **27.44 ±0.16** | **0.26 ±0.09** | **0.05 ±0.03** | **18.34 ±3.93** | **19.35 ±3.12** | **0.32 ±0.11** | **1.99 ±0.30** |
| | 25 50 | -1.38 -0.01 | 33.93 34.45 | 27.28 27.69 | 0.20 0.42 | 0.02 0.12 | 12.28 22.19 | 13.92 22.32 | 0.23 0.50 | 1.65 2.54 |

| | | | | | | | | | | |
|---|---|---|---|---|---|---|---|---|---|---|
| PSWw n=73 | **17** ±15 | **4.13** ±1.9 | **34.33** ±0.61 | **27.22** ±0.44 | **0.20** ±0.05 | **0.04** ±0.02 | **18.69** ±3.15 | **19.13** ±2.70 | **0.25** ±0.06 | **1.54** ±0.28 |
| | 1  50 | 0.37  8.14 | 32.17  34.89 | 25.80  27.70 | 0.12  0.34 | 0.01  0.09 | 11.51  24.96 | 13.56  24.87 | 0.15  0.40 | 0.96  2.63 |
| AAW n=9 | **76** ±76 | **1.69** ±0.2 | **34.72** ±0.09 | **27.77** ±0.08 | **0.25** ±0.06 | **0.05** ±0.02 | **17.72** ±2.81 | **18.28** ±2.42 | **0.28** ±0.07 | **1.64** ±0.38 |
| | 5  257 | 1.49  1.96 | 34.64  34.88 | 27.71  27.91 | 0.15  0.33 | 0.02  0.08 | 13.90  23.42 | 15.06  23.40 | 0.19  0.37 | 1.18  2.26 |
| IW/DW n=19 | **2175** ±604 | **-0.70** ±0.1 | **34.92** ±0.01 | **28.08** ±0.01 | **0.14** ±0.05 | **0.02** ±0.01 | **21.22** ±3.58 | **21.32** ±2.71 | **0.19** ±0.07 | **1.49** ±0.46 |
| | 794  2872 | -0.79  -0.15 | 34.91  34.93 | 28.06  28.10 | 0.09  0.27 | 0.01  0.06 | 13.32  27.90 | 15.57  26.59 | 0.12  0.44 | 1.03  2.46 |

Table 3. Yearly averaged descriptive statistics of selected CDOM optical properties from AREX 2013–2015.

| Year N | $\Theta$ [°C] | S | $a_{CDOM}(350)$ [m$^{-1}$] | $a_{CDOM}(443)$ [m$^{-1}$] | $S_{275-295}$ [µm$^{-1}$] | $S_{300-600}$ [µm$^{-1}$] | $a^*_{CDOM}(350)$ [m$^2$g$^{-1}$] | $SUVA_{254}$ [m$^2$ gC$^{-1}$] | DOC [µmol/l] | N | $Chla$ [mg/m$^3$] |
|---|---|---|---|---|---|---|---|---|---|---|---|
| 2013 79 | **4.69** ±1.77 | **34.62** ±0.63 | **0.30** ±0.12 | **0.06** ±0.05 | **15.39** ±3.24 | **17.94** ±2.68 | **0.32** ±0.11 | **1.79** ±0.66 | **80.69** ±24.46 | 71 | **0.87** ±1.13 |
| | -1.35  7.48 | 32.21  35.10 | 0.15  0.90 | 0.01  0.32 | 10.53  28.32 | 9.95  30.06 | 0.15  0.60 | 0.95  3.80 | 40.46  127.45 | | 0.07  8.83 |
| 2014 221 | **4.72** ±2.18 | **34.86** ±0.52 | **0.14** ±0.06 | **0.02** ±0.00 | **14.65** ±2.63 | **20.71** ±5.26 | **0.17** ±0.08 | **1.73** ±1.23 | **77.57** ±22.10 | 138 | **1.58** ±1.38 |
| | -0.91  7.45 | 31.29  35.09 | 0.04  0.34 | 0.10  0.02 | 10.51  26.04 | 10.83  42.26 | 0.05  0.59 | 0.56  9.23 | 40.28  131.70 | | 0.12  10.42 |
| 2015 263 | **4.04** ±2.23 | **34.78** ±0.45 | **0.19** ±0.05 | **0.03** ±0.02 | **19.26** ±2.91 | **19.64** ±2.44 | **0.23** ±0.06 | **1.47** ±0.30 | **67.64** ±6.50 | 142 | **1.39** ±0.83 |
| | -1.38  8.15 | 32.17  35.09 | 0.09  0.42 | 0.01  0.12 | 10.94  27.90 | 13.08  26.59 | 0.12  0.50 | 0.86  2.63 | 51.12  121.83 | | 0.14  3.70 |

*3.2. Optical properties of different water masses*

All measured salinity and temperature values are presented in the temperature–salinity (TS) diagram as a function of depth (Figure 3a) to visualize water masses sampled during AREX2013, AREX2014 and AREX2015 campaigns. Majority of measurement represented characteristics of AW that covered all depth ranges. The second water mass represented in our data set was low density PSWw ($\sigma_\theta \leq 27.7$ kg m$^{-3}$), which was observed above 50 m depth. The smallest fraction of data points belonged to PSW, which was aggregated in subsurface, 20–70 m depth range and AAW which was encountered within 50–100 m depth range (Figure 3a). To visualize the distribution of DOM properties within classified water masses we have chosen the fluorescence intensity of the marine humic–like DOM ($I_{CH1}$), fluorescence intensity of the protein–like DOM ($I_{CH3}$), and CDOM absorption $a_{CDOM}(350)$. The highest $I_{CH1}$ values were observed in PSW and lowest in PSWw (Figure 3b). Humic–like FDOM in AW was characterized with large dynamic range and both low (320 R.C.) and high values (>360 R.C.), were observed (Figure 3b). In case of $I_{CH3}$ the highest values were observed in PSW, PSWw mid depth (15-50m, what can be associated with chlorophyll a maximum) and in part of AW, which was separated from PSWw (upper part: T>0, $\sigma_\theta \leq 27.7$, S>34.9). The lowest $I_{CH3}$ values were observed in AW (lower part: $27.7 < \sigma_\theta \leq 27.97$) and in PSWw where $\sigma_\theta \leq 26.5$ (Figure 3c). There was a large variability and no consistent trends in distribution of $a_{CDOM}(350)$ values in different water masses in the study area, as shown in the TS diagram (Figure 3d). The distribution of fluorescence intensity of the terrestrial humic–like DOM ($I_{CH2}$) and SUVA$_{254}$ in the TS diagram is shown in the supplementary information (Figure S2).

a)

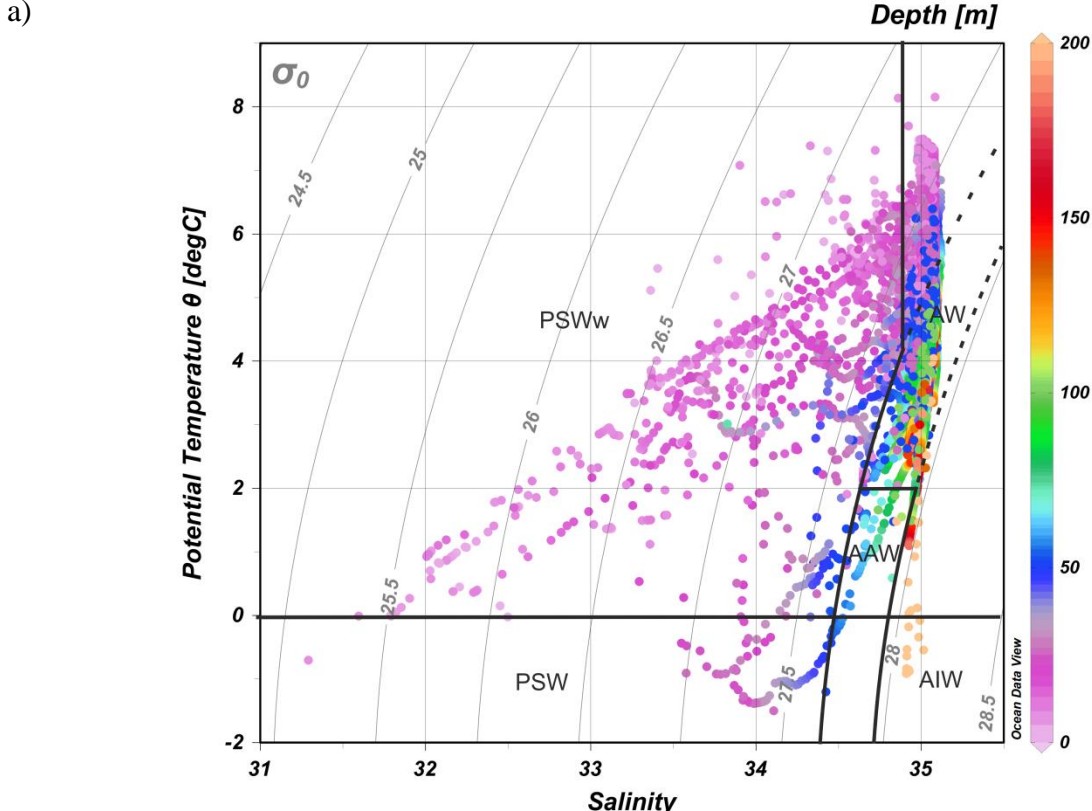

b)

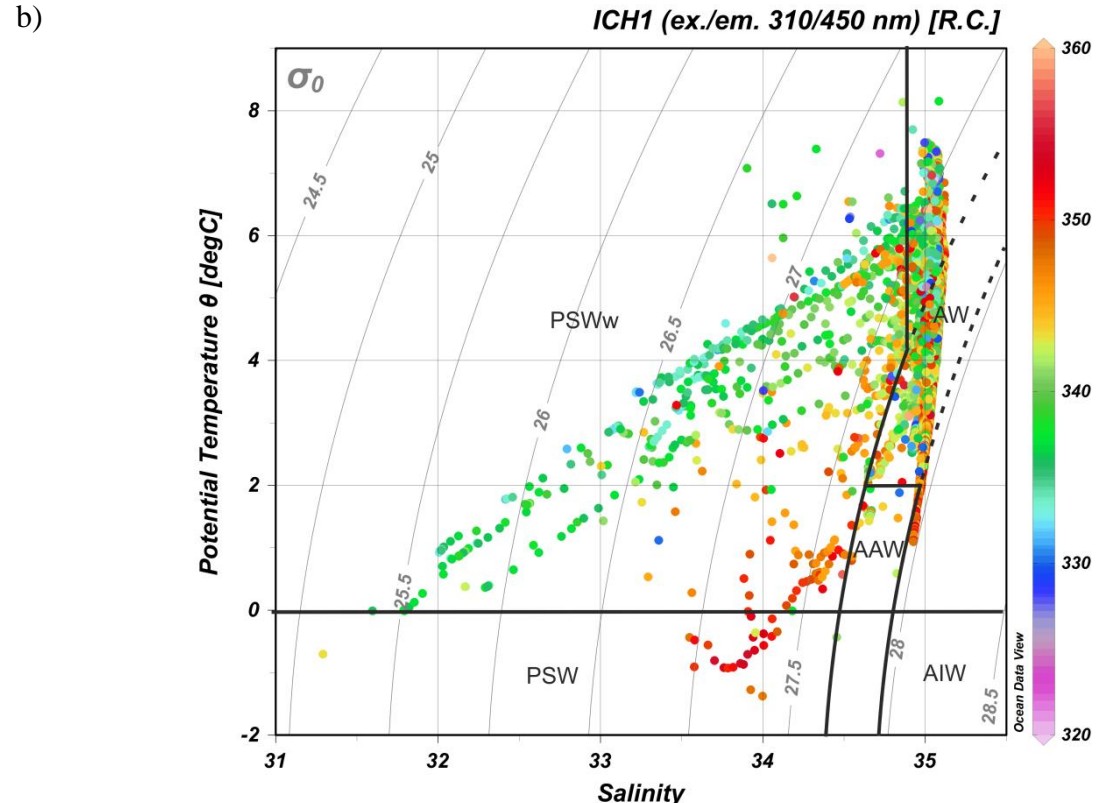

c)

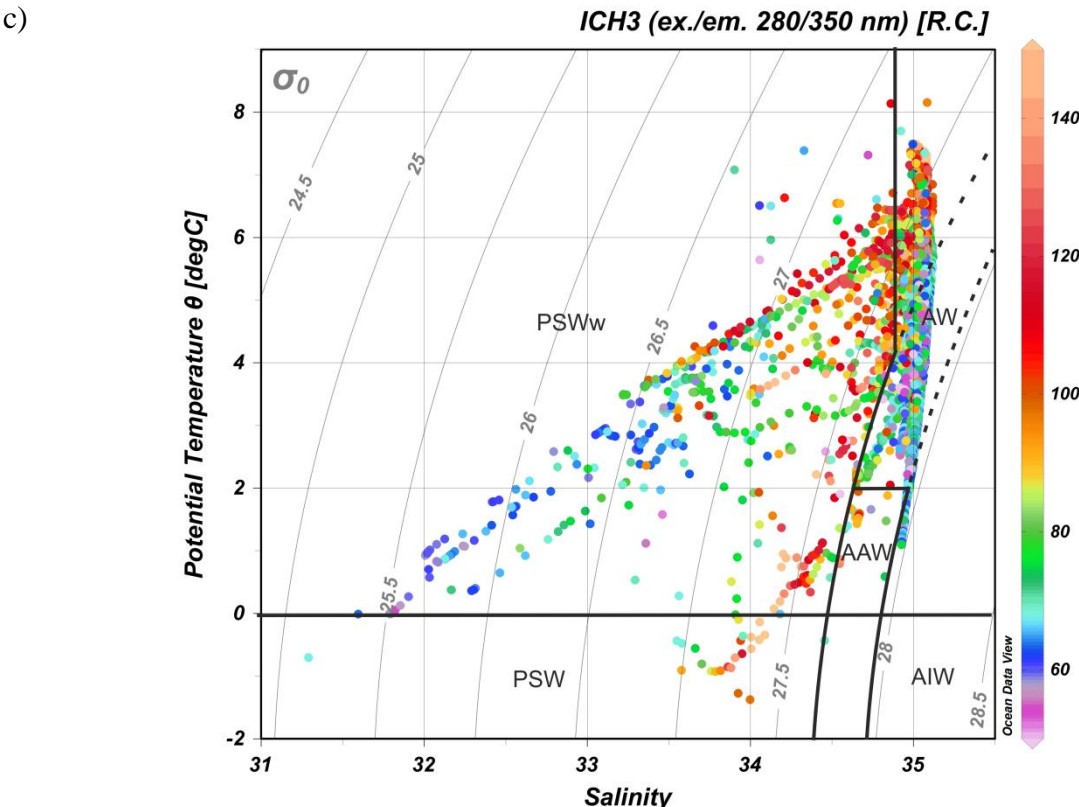

d)

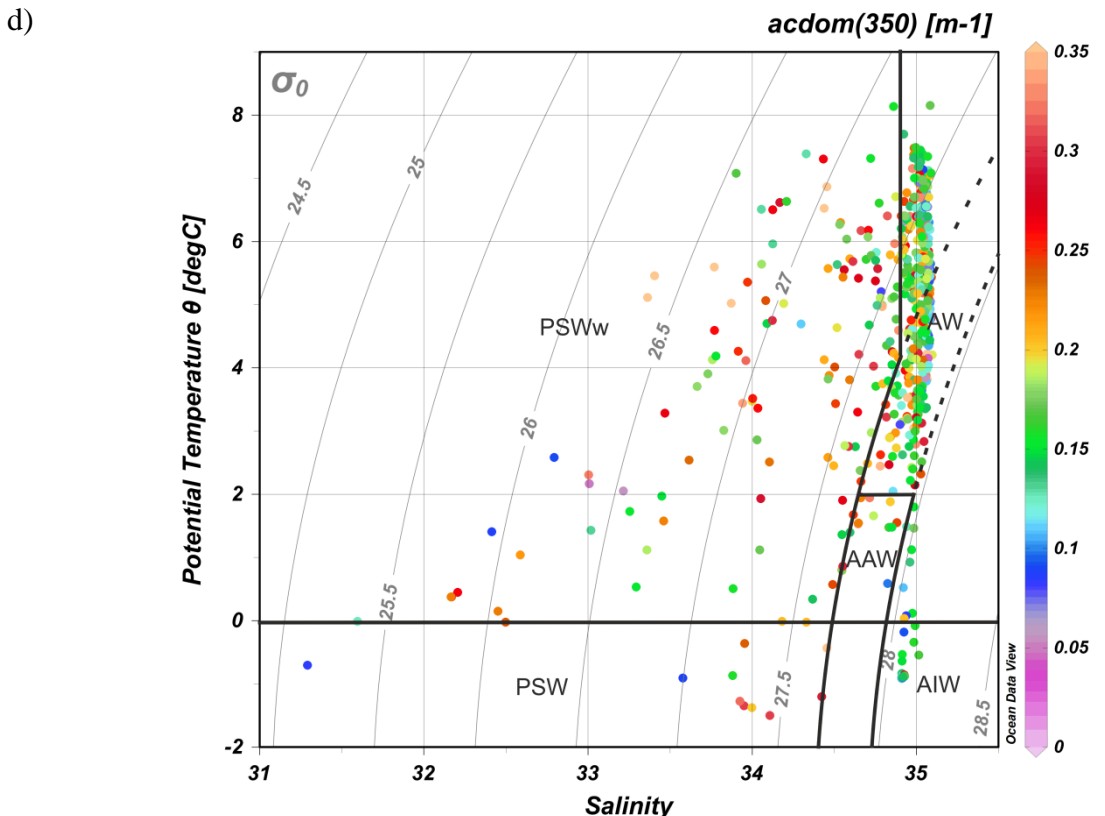

Figure 3: TS diagram of water mass distribution in the study area in 2013–2015. A)
colorbar represents depth [m]. B) colorbar represents humic–like fraction
fluorescence intensity $I_{CH1}$, (ex./em. 310/450 nm, [R.C.]). C) colorbar
represents protein–like fraction fluorescence intensity $I_{CH1}$, ( ex./em. 310/450
nm, [R.C.]). D) colorbar represents values of absorption coefficient at 350
nm, $a_{CDOM}(350)$ [m$^{-1}$]. The lower number of points in D) resulted from fewer
number of discrete water samples for determination of CDOM. Water
masses: AW (Atlantic Water), AAW (Arctic Atlantic Water), AIW (Arctic
Intermediate Water), PSW (Polar Surface Water), PSWw (Polar Surface
Water warm). Three areas noted as AW follow the three sets of conditions
that define AW (see Table S1).
*3.3. Vertical distribution of FDOM components*
The instrumental *in situ* synchronous IOP measurements enabled to resolve FDOM
distribution with better resolution, compared to coarser discrete water sampling of CDOM.
Representative vertical profiles of temperature, salinity, FDOM and chlorophyll *a*
fluorescence are shown in Figure 4. Differences in the vertical distribution of salinity and
temperature (Figure 4a,b) were observed in sampling stations located near the sea ice edge
(black stars), where a cold and fresher surface layer (typically 5–10 m deep; classified as
PSWw) was present. The salinity at stations located in the core of Atlantic waters (green
circles) and at the south–western Spitsbergen shelf (red circles) was uniform in the upper 100
m (Figure 4a,b). There was very little spatial and vertical variation in humic–like FDOM
($I_{CH1}$ and $I_{CH2}$). The only exception was the slightly higher, but still vertically homogenous
distribution, of humic–like FDOM observed at stations near the Spitsbergen coast in 2014.
(red dots; Figure 4c,d).
The vertical distribution of protein–like FDOM ($I_{CH3}$, Figure 4e) was very similar to
distribution of chlorophyll *a* fluorescence ($I_{FChla}$, Figure 4f) and total non–water absorption
coefficient at 676 nm ($a_{tot-w}(676)$, Figure 4g). All three parameters had a strong subsurface
maximum at the depth range between 10 and 30–40 m and similar spatial distribution. The
surface values for these three parameters were higher than values below the maximum (40 m)
for profiles in the AW (green and red symbols). Near the ice edge, however, stations were
characterized by lower values in the surface layer, comparable to the values below 40 m,
likely due to dilution of FDOM and *Chla* by sea–ice melt water at the very surface. The
$a_{tot-w}(676)$ vertical profiles in AW were different, with elevated values throughout the whole
upper layer (0–30 m depth), which dropped sharply to a background level below the
subsurface chlorophyll *a* maximum.

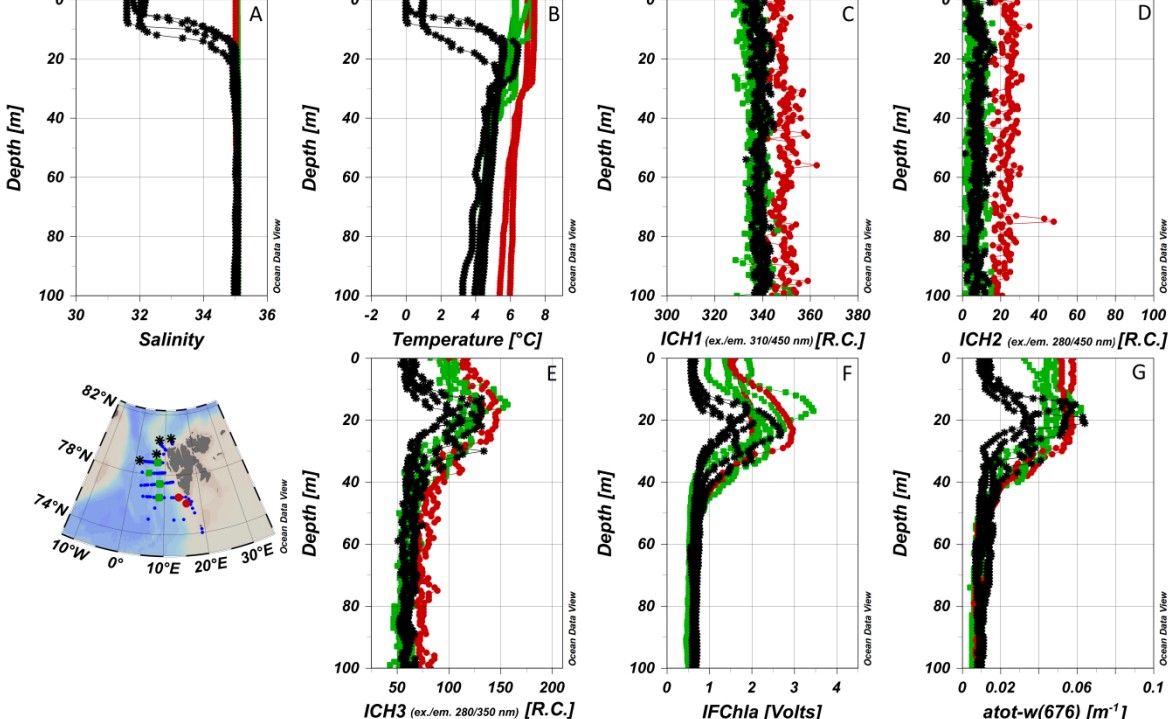


Figure 4: Vertical profiles of: salinity (A), temperature (B), different FDOM
components: marine humic–like FDOM ($I_{CH1}$, C), terrestrial humic–like
fraction of DOM ($I_{CH2}$, D), protein–like FDOM ($I_{CH3}$, E), chlorophyll *a*
fluorescence ($I_{FChla}$, F) and total non–water absorption coefficient at 676 nm
($a_{tot-w}(676)$, G) in 2014. Red dot, green square, black star symbols
correspond to vertical profiles obtained over the West Spitsbergen Shelf
(influenced by SC), in the core of the WSC, and near the ice edge (with a
presence of PSWw in the surface 0–20 m layer), respectively.
*3.4. Relationship between chlorophyll a and protein–like FDOM*
The qualitative correspondence between fluorescence of protein–like FDOM and
chlorophyll *a* fluorescence intensity (Figure 4) has been quantitatively confirmed by
regression analysis. A significant positive relationship between $I_{CH3}$ and $I_{FChla}$ was found in
both 2014 and 2015 ($R^2 = 0.65$, p<0.0001, n = 24490; Figure 5a). The relationship was
stronger in 2014 ($R^2 = 0.75$, p<0.0001, n = 17700, blue line in Figure 5a) when broader
influence of AW water was observed (Walczowski et al., 2017), than in 2015 ($R^2 = 0.45$,
p<0.0001, n = 7290, red line in Figure 5a).
The same relationship was confirmed using data from discrete water samples. A
statistically significant relationship between $I_{CH3}$ and *Chla* values was found in both years,
and the determination coefficient for combined data set was $R^2 = 0.36$ (p<0.0001 ) (Figure
5b). There was higher correlation observed between $I_{CH3}$ and *Chla* values in 2015 compared
to 2014 (Figure 5b). Higher dispersion between FDOM fluorescence intensity measured *in*
*situ* and chlorophyll *a* measured in water samples could be a result of the time lag between
instrumental measurements and water collection that reached up to 1.5 hours. The IOP
instruments deployment were usually done simultaneously with CTD down cast, while water
sample collection was performed during CTD rosette up cast, that was significantly delayed
especially at deep water stations (at sampling stations location with water depth >1000 m).
Observed higher protein–like FDOM values per chlorophyll *a* concentration unit could be
explained by phytoplankton physiological response due to higher water temperature observed
in 2014 and consequent more efficient extracellular DOM release. This physiological effect is
evident in relationships between chlorophyll a fluorescence and $a_{tot-w}(676)$. In 2014
phytoplankton were more fluorescent at the same absorption level (Figure S3).

a)

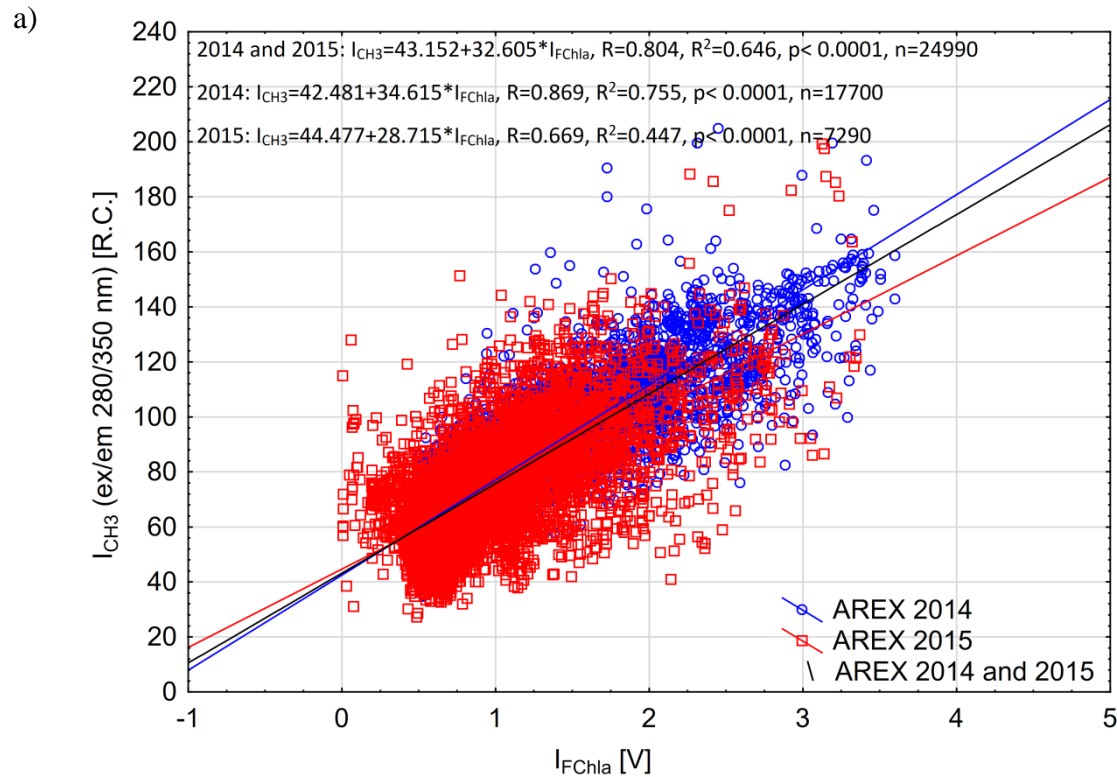

b)

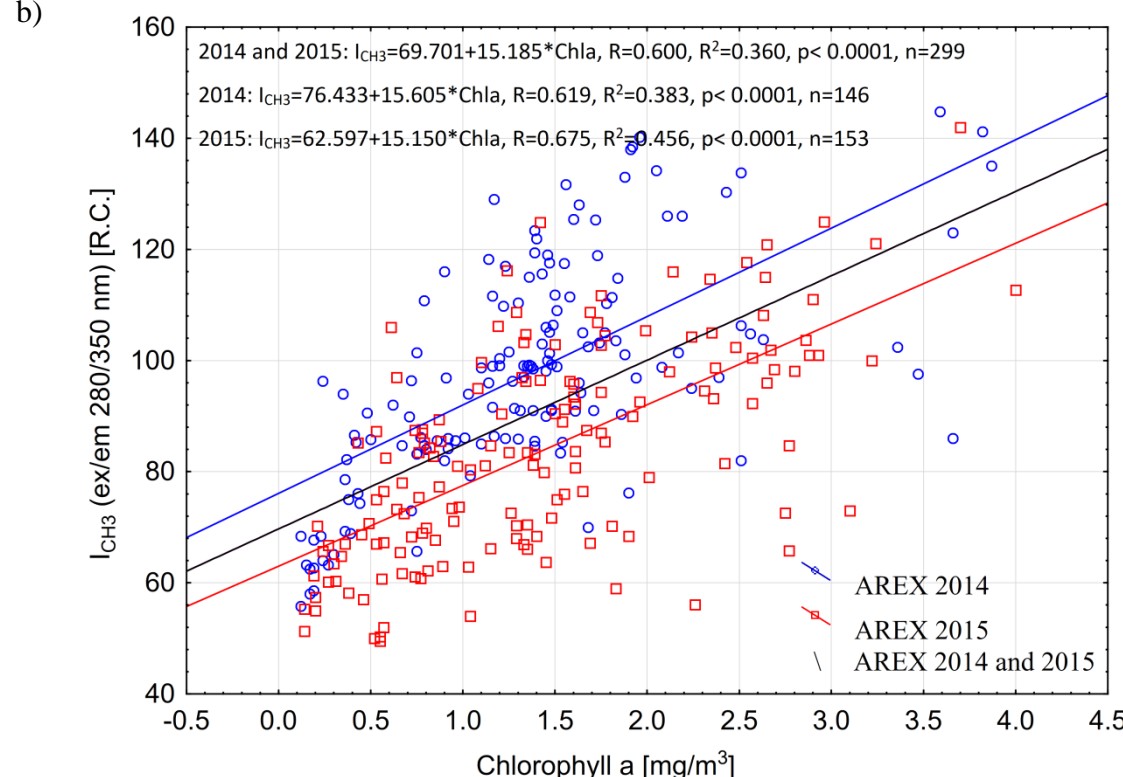

Figure 5. Relationship between chlorophyll a fluorescence ($I_{FChla}$) and fluorescence of the protein–like component ($I_{CH3}$) (a) and relationship between fluorescence of the protein–like component ($I_{CH3}$) and chlorophyll a concentration from discrete water samples (b) in the upper 200m of water column in 2014 and 2015. Set of linear regression functions, correlation coefficient (R), coefficient of determination ($R^2$), p–value and number of samples (n) are presented on Figure 5 on Figure 5.

## 4. Discussion

### 4.1. Variability and spectral properties of CDOM in the Nordic Seas

The highest CDOM absorption in the Arctic has been observed in coastal margins along Siberian Shelf in Laptev Seas, close to Lena River delta; $a_{CDOM}(440) = 2.97$ m$^{-1}$, salinity close to 0; (Gonçalves–Araujo et al., 2015) and in Laptev Sea Shelf Water at the surface; $a_{CDOM}(443) > 1$ m$^{-1}$, salinity <28, (Gonçalves–Araujo et al., 2018) and at the coast of Chukchi Sea and Southern Beaufort Sea influenced by riverine inputs of Yukon and Mackenzie Rivers; $a_{CDOM}(440) > 1$ m$^{-1}$, salinity <28, (Matsuoka et al., 2011, 2012; Bélanger

et al., 2013). Exceptionally high CDOM absorption has been also observed in the southern part of Hudson Bay near rivers outlets with $a_{CDOM}(355)>15$ m$^{-1}$, at salinity close to 0 (Granskog et al., 2007). Pavlov et al. (2016) reported $a_{CDOM}(350)$ of up to 10 m$^{-1}$ at salinity of 21 in surface waters of the White Sea. Terrestrial CDOM from Siberian Shelf has been diluted and $a_{CDOM}(440)$ decreased to ca. 0.12 m$^{-1}$, at salinities 32.6 (Gonçalves–Araujo et al., 2015) and transported further toward the Fram Strait by the Transpolar Drift being gradually diluted or removed (Stedmon et al., 2011; Granskog et al., 2012). In the Transpolar Drift and the central Arctic Ocean, CDOM absorption in surface waters was dominated by terrestrial sources with observed $a_{CDOM}(443)$ values varied between ~0.15 m$^{-1}$, at salinities close to +/- 27 (Lund–Hansen et al., 2015) and ~0.5 m$^{-1}$ at salinity range from 26.5 to 29.5 (Gonçalves-Araujo et al., 2018). Dilution also effectively decreased CDOM absorption in western Arctic Ocean, and average CDOM absorption in the Chukchi Sea and Beaufort Seas was $a_{CDOM}(440) = 0.046$ m$^{-1}$, at salinities > 32.3 (Matsuoka et al., 2011, 2012; Bélanger et al., 2013). The influence of transformed Atlantic Water generated in the Barents and Norwegian Sea had impacted on $a_{CDOM}(443)$ values in the Beaufort Gyre and Amundsen and Nansen basins, causing its decrease below 0.2 m$^{-1}$ as reported by Gonçalves–Araujo et al. (2018).

The reported lower range of $a_{CDOM}(350)$ observed in AW during AREX2014 (2014: 0.14±0.06 m$^{-1}$) is in good agreement with data from eastern part of Fram Strait at 79°N section reported by Granskog et al. (2012), Stedmon et al. (2015) and Pavlov et al. (2015) and with data reported by Hancke et al. (2014) south of the Polar Front in the Barents Sea. Kowalczuk et al. (2017) observed similar $a_{CDOM}(350)$ north of Svalbard. Higher values of CDOM absorption in AW observed in 2015 were within published variability range (Pavlov et al., 2015; Hancke et al., 2014; Kowalczuk et. al, 2017). Highest $a_{CDOM}(350)$ values in AW in 2013, 0.28±0.07 m$^{-1}$ (Table 2) were similar to Hancke et al. (2014) north of the Polar Front in the Barents Sea. Very low values of $a_{CDOM}(443)$ aligned with previous reports: in the core Atlantic waters in Greenland Sea measured during TARA Expedition in 2013 (Matsuoka et al., 2017), in the eastern Fram Strait (Pavlov et al., 2015) and in the Barents Sea (Hancke et al., 2014) and north of Svalbard (Kowalczuk et al., 2017). It should be underlined that data comparison could be biased by number of observations, as this study documented $a_{CDOM}(350)$ and $a_{CDOM}(443)$ statistics based on significantly higher number of samples and wider spatial coverage compared to the sources cited above.

The AW inflow with the WSC is an extension of NAC originating from the Atlantic
Ocean and CDOM absorption presented in this study were comparable with values found in
the North Atlantic Ocean (Kowalczuk et al., 2013; Kitidis et al., 2006). In contrast, values of
absorption coefficients were two times higher in Norwegian Coastal Waters which are
influenced by Lofoten Gyre, and presumably by terrestrial runoff as reported by Nima et al.
(2016).

Despite lower salinity and lower temperature CDOM optical properties in PSW in this
study did not differ significantly from AW in 2013 and 2015, and similar variability ranges of
CDOM properties were mention by Pavlov at el. (2017) north of Svalbard. Therefore, PW in
the eastern Fram Strait has not been advected from the central Arctic Ocean, as in the EGC
(Granskog et al., 2012; Pavlov et al., 2015), but rather it is a modified AW, strongly affected
by heat loss and diluted by sea–ice melt in the Barents Sea. Similar processes occur also on
North Spitsbergen Shelf, where PW was also found near the ice edge in surface waters diluted
and cooled by sea–ice melt.

According to Aas and Hokedal (1996) freshwater run–off from different sources
influence Svalbard waters and there is no universal relation between salinity and CDOM in
this area. Average values of $a_{CDOM}(350)$ in 2014 in PSW (Table 2) were similar with Arctic
Waters north of the Polar Front in Barents Sea described by Hancke et al. (2014) and slightly
higher than observed in this study in 2013 ($0.32\pm0.16$ m$^{-1}$) and 2015 ($0.26\pm0.09$ m$^{-1}$).
According to Hancke et al. (2014) the CDOM pool in the Barents Sea was predominantly of
marine origin, while several studies show terrestrial CDOM in the PW of EGC (Granskog et
al., 2012; Pavlov et al., 2015) and $a_{CDOM}(350)$ reported for PW in the EGC was significantly
higher, by factor 2, than values reported in this study around Svalbard.

CDOM absorption in WSC reported by Pavlov et al. (2015) and our observations
enabled to observe significant interannual variability of $a_{CDOM}(350)$ since 2009 until 2015.
The year to year changes in average $a_{CDOM}(350)$ may differ in AW as much as 200% (Table
2). We link these changes with intensity of AW transport to the West Spitsbergen Shelf
presented as spatially and vertically average salinity and temperature time series (Walczowski
et al., 2017). According to this study the average temperature north of 74°N was higher in
2009 than in 2010 that corresponded to lower $a_{CDOM}(350)$ in 2009 relative to 2010 (Pavlov et
al., 2015). Similarly in 2013, with highest CDOM absorption in our observations, the
temperature was lower than in 2014 and 2015 (Walczowski et al., 2017). The average salinity

of 35.05 reported in 2014 by Walczowski et al. (2017) was close to record high of 35.08 measured in the period 2000–2016. In 2014 we observed the lowest $a_{CDOM}(350)$ reported since 2009.

$S_{300–600}$ varied very little between water masses in a given season (Table 2), thus we assume that average seasonal values are representative for all water masses (Table 3). The largest variation of $S_{300–600}$ (Figure 6, Table 3) was observed in 2014, while the lowest variation of this parameter and a shift towards lower values was observed in 2013 and 2015. Spectral slope coefficient values (19.0±2.7 $\mu m^{-1}$) reported by Granskog et al. (2012) for AW across a section in eastern Fram Strait were very similar to those found during AREX2013 and AREX2015 (Table 2). Spectral slopes presented by Granskog et al. (2012), however, were calculated in broader spectral range 300–650 nm, while Hancke et al. (2014) calculated spectral slope coefficient in narrower spectral range of 350–550 nm. Recalculation of the spectral slope coefficient for our data set in the spectral range 300–650 nm, resulted in an average increase of $S$ by <1 $\mu m^{-1}$ relative to $S_{300–600}$. The spectral slope reported by Hancke et al. (2014) varied between seasons; values in May 2008 (16±4 $\mu m^{-1}$) were higher than those observed in August 2007 (14±4 $\mu m^{-1}$) but both were similar with values reported in this study. Although Hancke et al. (2014) calculated spectral slope coefficient for a narrower spectral range resulted consistently in lower spectral slope values by ~2 $\mu m^{-1}$ their values were within the range of $S_{300–600}$ in the current dataset. In the WSC the $S_{300–600}$ values were higher than those for surface waters north of Svalbard in winter–spring reported by Kowalczuk et al. (2017). Observations reported by Kowalczuk et al. (2017) were conducted earlier in the season and samples were collected below sea ice, so CDOM was less exposed to solar radiation and was potentially less affected by photobleaching. The highest $S_{300–600}$ were found during AREX2014 (20.71±5.26 $\mu m^{-1}$), when over 79% samples were classified as AW, what could be associated with photomineralization of DOM in aging sea water (Obernosterer and Benner, 2004).

*4.2. Identification of CDOM sources*

According to Stedmon and Markager (2001) the non–linear relationship between spectral slope $S_{300–600}$ and $a_{CDOM}(375)$ allows to differentiate between terrestrial (allochthonous) and marine (autochthonous) CDOM pools as well as assess changes in CDOM composition. This approach was validated by Granskog et al. (2012), who found that

CDOM samples taken in PW with high fractions of meteoric water (i.e. river water) in the
western part of Fram Strait were outside the Stedmon and Markager (2001) model limits for
marine CDOM. Increasing spectral slopes and decreasing CDOM absorption provides
information about degradation of autochthonous CDOM originated from marine environment
(Stedmon and Markager, 2001; Whitehead and Vernet, 2000). We found decreasing $S_{300-600}$
values with increasing CDOM absorption in all three years (Figure 6). This is similar to that
presented by Kowalczuk et al. (2006) in the Baltic Sea and Pavlov et al. (2014) in
Kongsfjorden, West Spitsbergen. In our study almost all data points are within the Stedmon
and Markager (2001) model limits (Figure 6), and suggests a dominant marine
(autochthonous) source of CDOM. The highest $S_{300-600}$ (>25 $\mu m^{-1}$) with very low CDOM
absorption (<0.075 $m^{-1}$) suggest a highly degraded CDOM pool in 2014. In contrast, lower
values of $S_{300-600}$ (<18 $\mu m^{-1}$) with higher absorption (>0.15 $m^{-1}$) could indicate freshly
produced CDOM. Lack of correlation between salinity and $a_{CDOM}(\lambda)$ was found here (not
shown) as by Hancke et al. (2014), which further suggests a marine origin of organic matter in
the study area.

There were some data points, measured in 2013 characterized by absorption

(>0.25 $m^{-1}$) and spectral slope ~18 $\mu m^{-1}$ that were outside the upper Stedmon and Markager
(2001) model limits. These points could bias the $S_{300-600}$ and $a_{CDOM}(375)$ relationship
derived for present data set, and suggest either more terrestrial contribution at high
$a_{CDOM}(375)$ from local sources or influence of the Polar Water in the western part of the
Fram Strait or recirculating modified AW. Slight increase of humic–like DOM fluorescence
($I_{CH1}$ and $I_{CH2}$), observed near the south–western Spitsbergen shelf (Figure 4), could indicate
a small local contribution from a terrestrial CDOM source.

The presumed molecular structure of marine autochthonous DOM is composed mainly

with low molecular weight aliphatic organic compound characterized with low saturation with
aromatic rings (Harvey et al., 1983). SUVA$_{254}$ defined by Weishaar et al. (2003) is related to
aromatic ring content within the mixture of water soluble organic DOM. Massicotte et al.
(2017) presented the global distribution of SUVA$_{254}$, and found that SUVA$_{254}$ decreased
sharply in the aquatic continuum from fresh (4.8 $m^2 gC^{-1}$) to oceanic waters (1.68 $m^2 gC^{-1}$).
SUVA$_{254}$ also decreases with increasing salinity, rapidly in the salinity range 0–8.7, remained
stable in salinity 8.7–26.8 and decreased slowly until salinity reached oceanic values, and
further remained at stable level of ca. 1.7 $m^2$ $gC^{-1}$ (Massicotte et al., 2017). $SUVA_{254}$ values
presented in this study (Table 2) were at the lower end of the global range, close to the
oceanic end member values. The highest average $SUVA_{254}$ values were found in PSWw in
2013 (1.95±0.60 $m^2$ $gC^{-1}$) and PSW in 2014 and 2015 (1.96±0.63 $m^2$ $gC^{-1}$ and 1.99±0.30
$m^2$ $gC^{-1}$, respectively) and lowest in PSW (1.31±0.28 $m^2$ $gC^{-1}$) and AW (1.41±0.24 $m^2$ $gC^{-1}$)
in 2013 and 2015, respectively. Pavlov et al. (2016) reported $SUVA_{254}$ values at salinity
>34.3 in the southern Barents Sea waters in the range 1.3–1.8 $m^2$ $gC^{-1}$, which agree well with
our findings. The $SUVA_{254}$ values observed in the Siberian Shelf at salinity >30 varied
between 1.25–2.3 $m^2$ $gC^{-1}$, (Gonçalves–Araujo et al., 2015). Low $SUVA_{254}$ values suggested,
overall low saturation of CDOM with aromatic rings, which supports hypothesis on
predominantly autochthonous CDOM origin and minor influence by terrestrial DOM in the
Nordic Seas, with hydrography dominated by AW inflow.

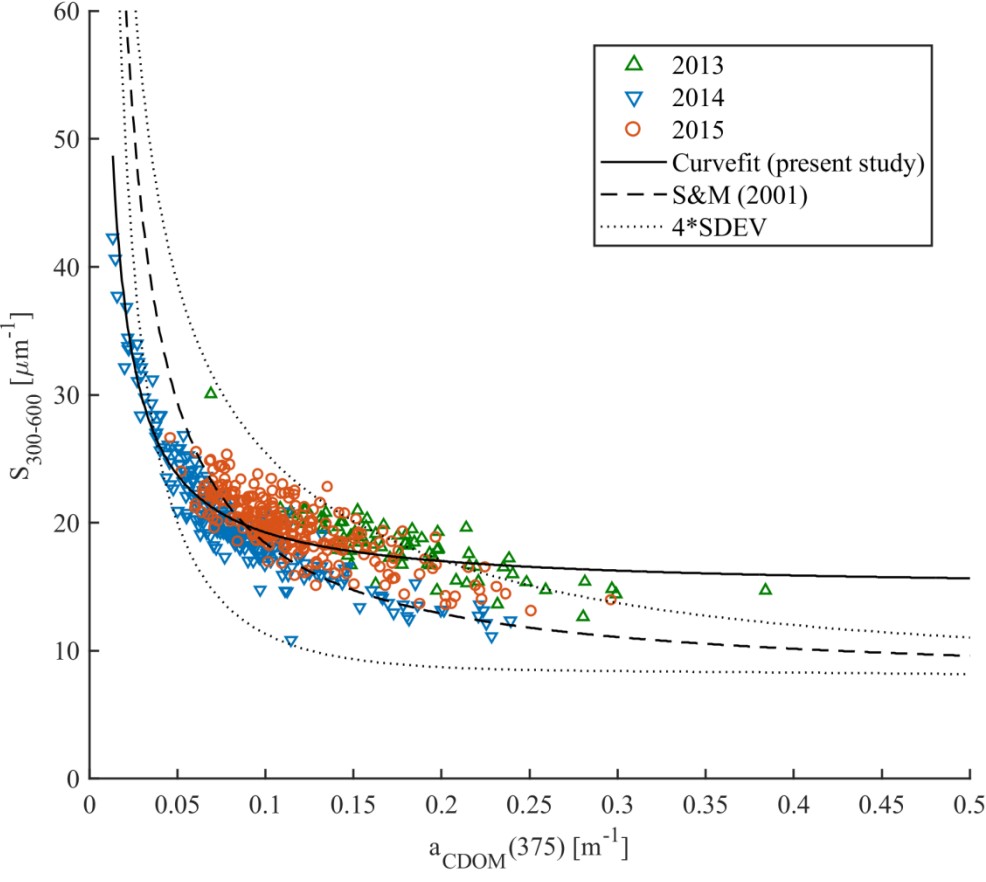



Figure 6: Spectral slope $S_{300-600}$ vs. $a_{CDOM}(375)$ in samples from 2013 (green
triangles), 2014 (blue reversed triangles), 2015 (red circles). Stedmon and

Markager (2001) model (dashed line) with model limits (±4 standard

deviation times the precision of the S–estimate; dotted line) adopted from

equation: $S=7.4+11/a_{CDOM}(375)$. Solid line represents the modeled nonlinear

fit for the present study data set.

*4.3 Relationship between CDOM absorption and DOC*

The significant amount of DOC in the Arctic Ocean mainly originates from riverine

inflow and permafrost thaw (Stedmon et al., 2011; Amon et al., 2012; Spencer et al., 2015).
The riverine input can be monitored by optical methods with absorption, fluorescence or
remote sensing measurements (Spencer et al., 2012; Walker et al., 2013; Fichot et al., 2013;
Mann et al., 2016). The largest DOC concentrations were found in Siberian rivers: e.g. Lena –
1300 μmol/l, Yenisey – 842 μmol/l, Ob – 950 μmol/l, and was lower in North American
Rivers: Yukon – 816 μmol/l and McKenzie – 648 μmol/l (Amon et al., 2012; Mann et al.,
2016). Both CDOM and DOC in coastal areas in the Arctic Ocean show an inverse
relationship with salinity (Amon et al., 2012) and very good correlation between CDOM
absorption and DOC has been reported for regions influenced by riverine input (Matsuoka et
al., 2012; 2013; Gonçalves–Araujo et al., 2015; Pavlov et al., 2016; Mann et al., 2016). The
DOC concentration observed by Amon et al. (2003) in the EGC in the western part of Fram
Strait and in Denmark Strait was considerably lower and ranged from 76 μmol/l in PSW to 55
μmol/l in AW. Amon et al. (2003) found a weak inverse relationship between DOC and
salinity in the Nordic Seas and a weak correlation between DOC and CDOM fluorescence.
The DOC concentration reported in this study in AW dominated eastern part of Fram Strait
was similar to that reported by Amon et al. (2003) in the EGC, but lower than found in
Barents Sea waters entering the White Sea at salinities close to 34.9 (Pavlov et al., 2016). The
DOC concentration in open Laptev Sea was over 100 μmol/l as reported by Gonçalves–
Araujo et al. (2015). We observed a very weak correlation between DOC concentration and
$a_{CDOM}(350)$ (Figure 7). That could be explained by low number of samples influenced by
terrestrial humic substances in our data, that have elevated $a_{CDOM}(350)$, DOC and lower
salinity. Additionally our data were at the lower range of globally observed distribution of
DOC and $a_{CDOM}(350)$, where the relationship is characterized by large uncertainty
(Massicotte et al., 2017).

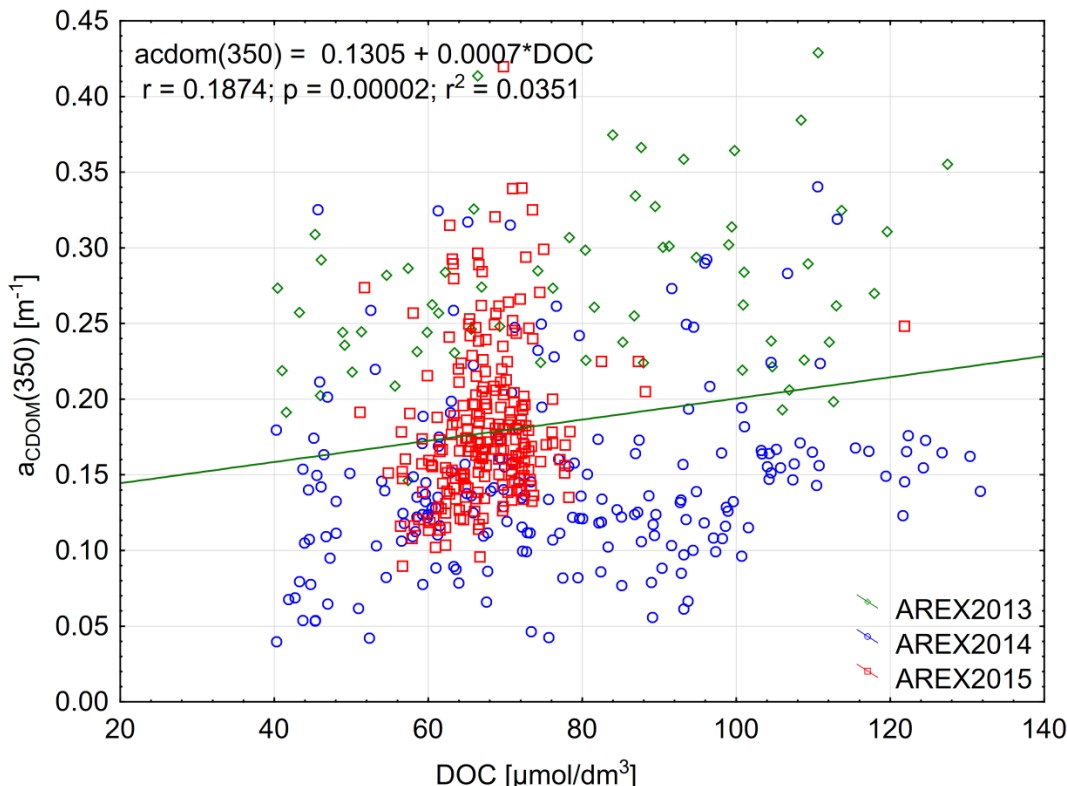

Figure 7. Relationship between $a_{CDOM}(350)$ and DOC and linear relationship between those parameters in 2013–2015.

The relationship between the carbon–specific CDOM absorption coefficient $a^*_{CDOM}(350)$ and $S_{275-295}$ was another approach suggested by Fichot and Benner (2011, 2012) in the Gulf of Mexico to trace the influence of terrigenous DOC in coastal margins, and to estimate DOC from optical measurements. We did not observe a significant relationship between $a^*_{CDOM}(350)$ and $S_{275-295}$ (not shown). However, $a^*_{CDOM}(350)$ as a function of $S_{300-600}$ showed much more promise (Figure 8). This could be potentially applied for DOC estimations from CDOM absorption measurements in Nordic Seas.

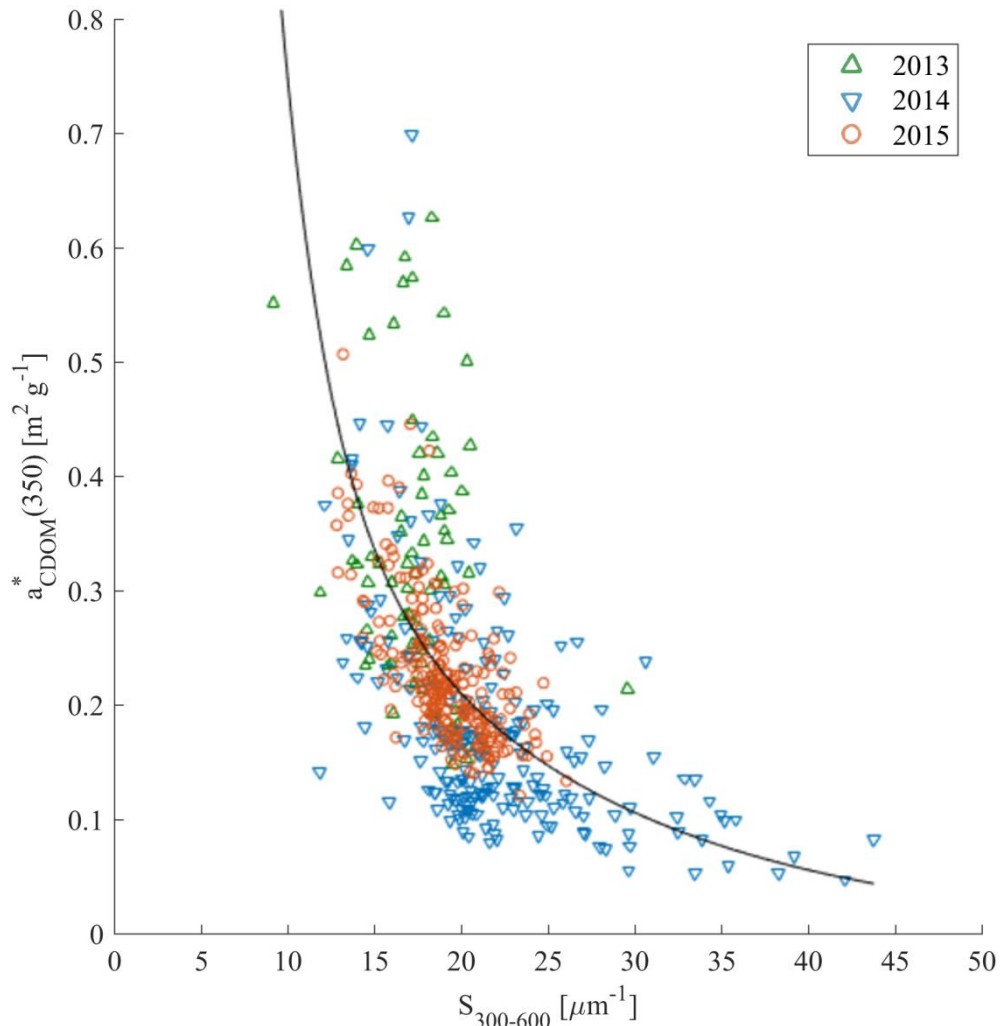

Figure 8. $a^*_{CDOM}(350)$ against $S_{300–600}$ in samples from 2013 (green triangles), 2014 (blue reversed triangles), 2015 (red circles). Non–linear fitting function between those parameters was adopted from Fichot and Benner (2012): $y = e^{(b_1 - b_2 x)} + e^{(b_3 - b_4 x)}$; regression coefficients $b_1 = 0.0027$; $b_2 = 73.31$, $b_3 = 1.29$; $b_4 = -91.39$, were estimated with Matlab curve fitting toolbox, with determination coefficient $R^2 = 0.38$, $n = 525$.

## 4.4. Distribution of FDOM components in the ocean and their dependence from allochthonous and autochthonous sources

The distribution pattern of main FDOM components with depth in the global oceans' biogeochemical provinces is significantly different for humic–like and protein–like FDOM (Stedmon and Nelson, 2015, Catalá et al., 2016). The intensity of humic–like FDOM fraction is usually higher close to continental margins and significantly depleted in the centers of

subtropical gyres (Murphy et al., 2008; Jørgensen et al., 2011; Kowalczuk et al., 2013). The fluorescence of humic–like DOM fractions is low in the surface layer and is rapidly increasing with depth reaching a constant high level below 200 m depth. Protein–like FDOM fluorescence intensity usually increases toward the open ocean and the highest intensity is observed in the surface waters, rapidly decreasing with depth, reaching constant low level below epipelagic layer (Jørgensen et al., 2011; Kowalczuk et al., 2013; Catalá et al., 2016). Such profiles indicate that amino acid–like DOM is linked to surface water production. Catalá et al. (2016) demonstrated that the global depth distribution tryptophan–like FDOM component has a local maximum associated with a chlorophyll *a* maximum. The linkage between protein–like components with chlorophyll *a* concentration shown qualitatively in the global ocean by Stedmon and Nelson (2015) and Catalá et al. (2016) was previously confirmed quantitatively in the mesocosm studies e.g. Romera–Castillo et al. (2010), which indicated that phytoplankton–excreted tryptophan like fluorophores, and tryptophan–like components concentration has been related to primary production (Brym et al., 2014). *In situ* quantitative correlation between chlorophyll *a* concentrations and fluorescence intensity of protein–like FDOM fraction has been observed and documented recently. Yamashita et al. (2017) reported significant positive correlation of tryptophan–like component and *Chla* (r = 0.53, p<0.001) in the surface waters of the Pacific Ocean. Yamashita et al. (2017) found also spatial coupling between the tryptophan–like component and chlorophyll *a* concentration which was strongest in Bering Sea. Study by Loginova et al. (2016) from Peruvian upwelling system also reported positively correlated chlorophyll *a* concentration and protein–like component ($R^2$ =0 .40, p<0.001).

The distribution of fluorescence intensity of main FDOM components in the Nordic Seas, dominated by warm water of Atlantic origin followed the general trends observed globally. The highest FDOM intensity, especially of humic–like components was observed close to continental margins, in the vicinity of major rivers outflows. Para et al. (2013) observed significant inverse trends of humic–like FDOM components with salinity in the Canadian shelf of the Beaufort Sea close to McKenzie River outflow. Similar observations were documented by Gonçalves–Araujo et al. (2015) in the Lena River delta at Siberian Shelf and by Pavlov at. al. (2016) near the Northern Dvina River outlet in the White Sea. The impact of humic–like FDOM component on DOM composition decreased with increased distance from fresh water sources and increased salinity, where the protein–like FDOM fraction became dominant e.g. outside of McKenzie River plume in Beaufort Sea (Para et al.,

2013) and in the White Sea (Pavlov et al., 2016). In the Fram Strait the distribution of humic–like fluorescence (Ex/Em = 340/420 nm) observed by Amon et al. (2003) in the Fram and Denmark Strait was related to large scale water masses distribution in Nordic Seas and was characterized with elevated values of FDOM intensity in the western part of Fram Strait that was under influence of EGC, and low FDOM intensity and uniformly distributed with depth FDOM intensity in the core of AW in its eastern part. The FDOM distribution in AW shown by Amon et al. (2003) corresponded well to vertical profiles of $I_{CH1}$ and $I_{CH2}$ in AW, shown on Figure 4. This was also in a good agreement with CDOM distribution in the Fram Strait (Granskog et al., 2012; Pavlov et al., 2015) and FDOM humic–like fraction (Ex/Em = 280/450 nm) distribution presented by (Granskog et al., 2015b). Humic–like fraction of DOM in the Eastern Fram Strait is more than 10 times lower compared to PW in EGC (Granskog et al., 2015b). A layer of 20 m deep of less saline water diluted by sea–ice melt characterized by significantly lower humic–like FDOM intensity was overlying the PW water with high FDOM intensity in EGC (Granskog et al., 2015b).

*In situ* fluorometry provided an opportunity to study FDOM distribution in greater detail and commercially available FDOM fluorometers are usually built to detect humic substances (Amon et al., 2003; Belzile et al., 2006; Kowalczuk et al., 2010; Aiken et al., 2011; Loginova et al., 2016). In this study we measured simultaneously three different FDOM components, and the most interesting feature observed with use of this new instrument was very significant spatial coupling between $I_{CH3}$ and $I_{FChla}$. Similarities in vertical distribution of protein–like FDOM, $I_{CH3}$ and stimulated chlorophyll *a* fluorescence intensity, $I_{FChla}$ and total non–water absorption coefficient at 676 nm, $a_{tot–w}(676)$ implied quantitative interrelation between those parameters and same dominant factor controlling these parameters in time and space. We found a significant positive correlation ($R^2 = 0.65$, p<0.0001) between $I_{CH3}$ and $I_{FChla}$ (Figure 5a) which suggests that production of protein–like FDOM is closely related with spatial and temporal phytoplankton dynamics. Additionally a statistically significant dependence of $I_{CH3}$ and *Chla* concentration from water samples indicated that phytoplankton biomass is an important source of protein–like FDOM.

**5. Conclusions**

We observed significant interannual variation of CDOM optical properties in the Nordic Seas. It is likely that these year to year changes in CDOM absorption coefficient and spectral slope coefficient were related to intensity of AW inflow to Nordic Seas. According to

Walczowski et al. (2017) there was very strong interannual variability in AW inflow overlaid
on the long–term increasing trend. CDOM absorption decreased and spectral slope coefficient
increased during years when increase of temperature was observed for Atlantic Waters (AW)
(Walczowski et al., 2017), e.g. in 2009 (Pavlov et al., 2015) and in 2014 (this study).
Decrease of AW temperature was accompanied by mutual increase of $a_{CDOM}(350)$ and
decrease of $S_{300-600}$, e.g.: in 2010 (Pavlov et al., 2015) and in 2013 and 2015 (this study). We
surmise that during less intense inflow of AW to Nordic Seas a higher proportion of PW is
transported with ESC and SC to eastern part of Fram Strait contributing to increase of CDOM
in West Spitsbergen Shelf waters.
*In situ* observations with use of a 3–channel fluorometer coupled with other optical
instruments enabled to show a significant correlation between protein–like FDOM and
chlorophyll *a* in the Nordic Seas. Quantitative dependence between protein–like FDOM
($I_{CH3}$) and chlorophyll *a* fluorescence ($I_{FChla}$) and between protein–like FDOM ($I_{CH3}$) and
total non–water absorption coefficient at 676 nm ($a_{tot-w}(676)$) based on direct *in situ*
observations clearly indicated that phytoplankton biomass is the primary source of low
molecular weight DOM fraction in Nordic Seas influenced by warm Atlantic waters. This
highlights the role of phytoplankton dynamics as an important factor controlling
FDOM/CDOM. Freshly produced protein–like FDOM fraction did not contribute to
CDOM/FDOM optical properties observed in visible spectral range as its fluorescence
excitation (absorption) and emission characteristics were located in the ultraviolet spectral
range. Observed variability of spectral indices ($a*_{CDOM}(350)$, $SUVA_{254}$, $S_{300-600}$) values
suggest that CDOM/FDOM in the Nordic Seas has an autochthonous origin. Yet, further
investigation of the DOM transformations processes from labile freshly produced protein–like
DOM fractions to more complex organic molecules is needed to better understand the
CDOM/FDOM dynamics in the Nordic Seas. Typically humic–like FDOM was found in low
concentrations in the study area, showcasing the limited terrestrial influence, in contrast to
e.g. the East Greenland Current (Gonçalves–Araujo et al., 2016).
Dissolved organic carbon (DOC) was weakly correlated with $a_{CDOM}(350)$ in the study
area, likely due to limited terrestrial influence, and $a_{CDOM}(350)$ shows no promise to be used
as a tool to predict DOC. The same was the case for spectral slope at short wavelengths ($S_{275-}$
$_{295}$), proven earlier to work for near–shore environs (Fichot and Benner, 2011, 2012). On the
other hand there was a significant inverse non–linear relationship of CDOM specific DOC
absorption ($a^*_{CDOM}(350)$) with spectral slope at a broader spectral range ($S_{300-600}$). This
relationship provides a potential for indirect estimates of DOC with use of optical
measurements in this region.
**Acknowledgements:** We thank the crew of RV Oceania and colleagues for the help onboard.
This work was supported by the Polish–Norwegian Research Programme operated by the
National Centre for Research and Development under the Norwegian Financial Mechanism
2009–2014 in the frame of Project Contract Pol–Nor/197511/40/2013, CDOM–HEAT. This
work was partially financed from the funds of the Leading National Research Centre
(KNOW) received by the Centre for Polar Studies for the period 2014–2018. MAG was
supported by the Centre for Ice, Climate and Ecosystems (ICE) at the Norwegian Polar
Institute, and AKP by the Research Council of Norway through the STASIS project
(221961/F20).

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
