# Peer review of "Characteristics of Chromophoric and Fluorescent Dissolved"

_Ocean Science, 2017_

## Referee Comment (RC1) · Anonymous Referee #1 · 8 Feb 2018

Abstract 1) page 2 line 45 why you concluded that phytoplankton is the main source of protein-like fluorescence based on a r2=0.36?

2) page 3, line 51. how did you arrive to this conclusion? of the Arctic Ocean (Arrigo et al., 2008), which could potentially contribute to increased production of autochthonous (marine) dissolved organic matter (DOM). what about the ice algae? they will disappear and they also contribute to CDOM

3) page 3 line 65, which percentage to carbon budget? DOM presents a considerable role in the carbon budget of...

4) line 71. please add Pegau reference to this list Hill, 2008; Granskog et al., 2007,

2015).

5) line 73 sorry this is not conclusive. UV can also produce radicals after interacting with CDOM resulting in more toxic and damaging effects! and preserves marine ecosystem from harmful ultraviolet radiation

6 )line 78, what fraction of DOM is CDOM? what fraction of CDOM is FCDOM?

7) page 4 line 100, upstream you meant? changes associated with CDOM in the areas downstream of the Atlantic Water inflow region

8) page 8 line 209, S between 300 and 600 nm line 218 why additional acdom375 and acdom443 when actually the range is 300 and 600 nm? line 224 why to use micron-1 use nm-1

9)page 9 equation 3, why the use of spectrophometry for chl? this is an old technique that has a larger error and is less sensitive than fluorometry or HPLC. What is the error of this emthod? Did you compare this method with HPLC or fluorometry non-acidification technique?

10) page 9 equation 4, I disagree. You cannot mix apples with bananas. DOC is not DOM unless you estimate DOM based on DOC with a curve or factor., same for equation 5 line 305, how did you calculate the offset of wetstar-3 fluorescence measurements? reference with respect to nanopure? constant temperature?

11) S slope without units?

12) figure 3 is hard to interpret due to the vertical variability of properties

13) many questions but fewer explanations or explanation attempts. beyond sampling aliasing, Why not acdom350 not well related to DOC? why not links between s275-295 and DOC? what is the linkage between particulate iron and absorption slopes?

---

## Referee Comment (RC2) · Anonymous Referee #2 · 7 Mar 2018

The manuscript by Makarewicz and colleagues presents combined results from three summer expeditions to the Nordic Seas, where they investigated the characteristics of both colored and fluorescent dissolved organic matter and their correlations to hydrographic variability. The dataset from the under sampled Nordic Seas presented in this study is interesting and of great value providing new and useful information on CDOM/FDOM in the study region. I suggest this manuscript for major revision before acceptance for publication in Ocean Science.

GENERAL COMMENTS

- The manuscript presents three major objectives in the introduction (Lines 113-118),

however the discussion as well as the abstract and conclusions do not clearly present the answer to the questions that the authors have raised. The authors may need to address those points accordingly through the manuscript. For instance, there is no clear distinction between large and regional circulation processes (objectives 1 and 2) with presenting and/or discussing the findings in the MS. The authors also state in their objective 3 that the impacts of phytoplankton on CDOM distribution (apart from FDOM), but only results concerning the impacts on the FDOM fraction are presented. - Several figures (e.g. Figs. 2, 3, 4) need to have their labels enlarged. - The authors mention several times through the text that some correlations or comparisons among years/expeditions were significant. However, the p-values are not given. Authors must present the p-values and also indicate the tests that have been applied. - The discussion section is superficial. The authors present results from other studies (like a review) and establish comparisons between their results and previous studies. However, no proper discussion regarding the results presented in the MS was performed. - Although the manuscript is generally well written and easy to follow, there are several grammar issues through the manuscript and I therefore suggest this manuscript to be revised by an English native speaker before submitting the revised version it.

SPECIFIC COMMENTS

L. 83: what do the authors mean by efficient? Please, be more specific, otherwise the reader can read interpret it as the other methods are not efficient.

L. 93-94: the contrasting optically properties of AW and PW are also with respect to the FDOM fraction (see Jørgensen et al 2014; Gonçalves-Araujo et al 2016). Additionally, such a contrast in CDOM properties is also highlighted in Stedmon et al 2015.

L. 94-97: missing references

L. 100: would be interesting to shown how CDOM/FDOM measurements can be used as a proxy for tracing other parameters (e.g., freshwater, DOC, phytoplankton primary production, etc).

L. 102-118: There is a recent paper that may be useful for the authors: Gonçalves-Araujo et al 2018.

L. 110-113: please consider re-writing the sentence. It is missing a verb.

L. 139: consider adding the (Bjørnøya Current) BC to figure 1.

L. 171-172: the water sample collected below the Chl-a maximum. Authors can provide more information regarding it. Was it within the pycnocline? Or at the bottom of the deep Chla maximum layer? How was the sampling depth determined?

L. 178-189: have the authors performed any test to check whether the differences in storage affected their results?

L. 250: was is the meaning of CRM?

L. 303: Sagan et al 2017. Authors should avoid citing unpublished work.

L. 315-317: It is worth to mention that the referred water masses are found within the "upper layer", given that no water mass from deep layers was presented.

L. 324: the meaning of the PSWw (as well as its thermohaline characteristic intervals) was not presented. Additionally, and for consistency. . . the PSWw is mostly referred in the literature as ASW (e.g. Pavlov et al 2015 and Gonçalves-Araujo et al 2016).

L. 339: I suggest the authors to add some specific words to the title that would better reflect the results presented in this sub-section: spatial variability and hydrography.

L. 340-357: please consider including the standard deviation or standard error to the averaged values presented through the text.

L. 353: 200 m water layer. The indication for meter is missing.

L. 358: what do the authors mean by distinct? Please be more specific.

L. 371-372: How do the authors explain the higher CDOM in PSWw compared to PSW? Shouldn't the DOM signal in PSWw be diluted by sea-ice melt waters and therefore,

lower than the DOM observed in the core of PSW as highlighted in Pavlov et al 2015, Stedmon et al 2015, Granskog et al 2012, Gonçalves-Araujo et al 2016, etc.?

L. 378-379: highest aCDOM(350) found within PSW and AAW in 2015. . . are those values for each water mass significantly similar? How do the authors explain such a high CDOM content associated to the AAW?

L. 380-382: Could this represent a consistency in the DOM content (fingerprint) with respect to each water mass?

L. 386-391: Since SUVA254 and a350* are both normalized by the same value (DOC concentration), what do the differences between the two parameters mean?

L. 396-397: is that decrease significant?

L. 396-399: what was the correlation between a350 and Chla? If autochthonous DOM is dominant in the sampling region, there should be a good correlation between the two parameters, right? (or at least for AW, with little continental/Arctic influence)

Fig 3b: It is clear that the Ich1 had the highest fluorescence intensity values in R.C., but, since the results of this MS are more focused on the Ich3 (autochthonous), I suggest the authors to present the Ich3 in this figure as well.

L. 469-470: "The relationship was more significant in 2014. . .". It looks like that the significance is the same for both years. Did the authors want to say that the relationship was stronger (given differences in r2)?

Fig 5a: Looking at the legend it is not clear what the black line means. Is it the regression curve for the combined dataset from 2014 and 2015 or was it the difference between them? I also think that the information displayed in Table 4 can be easily incorporated to Fig 5.

L. 508: At what salinity did Grankog et al 2007 found acdom(355) greater than 15m-1? It would make it easier to establish comparison to the other studies.

L. 502-518: There is a recent paper from Gonçalves-Araujo et al 2018 that shows aCDOM values for the central Arctic and surroundings that the authors may want to look at.

L. 521: do the values also agree with results published in Stedmon et al 2015?

L. 541: "... and similar features of CDOM properties were..." - What do you mean by similar features of CDOM properties?

L. 549: What do the authors mean by "Statistical distribution"?

L. 553: How did the authors infer that the waters are from different origins based on their dataset?

L. 556: Gonçalves-Araujo et al 2016 has studied FDOM and not CDOM as mentioned in the text.

L. 565-566: Have the authors tried to look for the a350 vs. temperature correlations?

L. 571-572: where did the authors take this information from?? Was it from the literature (then it is missing a citation) or from their dataset (then I would like to know how the data supports such an affirmation)?

L. 573: "S300-600 varied little between water masses in a given season..." Have you tested it for significance?

L. 612: "... could indicate freshly produced CDOM." . At what depth was this found? Have you looked at the correlation between S300-600 and Chla? Is there any correlation for the high Chla samples?

L. 611-614: This is an important discussion of the main results and could be discussed more in deep.

L. 617-620: Not clear whether the authors have checked that or it is only an assumption.

L. 623-625: Not clear what the authors wanted to state in that sentence.

L. 654: Authors may want to look at the results from Walker et al 2013 for comparison.

L. 682-683: Why do the authors think the a350* vs. S300-600 behaves like this? Was the correlation also strong (and significant) for each year analyzed separately? It does not look like that (specially for 2013) when looking at Fig. 8.

L. 695-696: significant differences for humic-like and PTN-like FDOM. . . With respect to what were these differences?

L. 699-701: How do the authors explain such behavior?

L. 702-705: not clear if the authors are presenting their own results or results from other studies.

L. 722-723: It is not completely clear based on the results presented here. The authors may want to rephrase that sentence. The authors state that the PTN-like FDOM is the dominant fraction at high salinity. However, studies have shown that the humic-like (visible fluorescence) can be the dominant signal at sal>34 in the Arctic Ocean.

L. 732-734: This sentence does not help the discussion and could be easily removed.

L. 742: The paper from Gonçalves-Araujo et al 2016 does not show CDOM results.

L. 756: ". . . and same dominant factor controlling these parameters in time and space." How do your results support that statement?

L. 762-763: Could the authors identify what FDOM fractions are produced through each of the mentioned processes (e.g. phytoplkt extracellular release, phytoplkt degradation or lysis)??
* * *

---

## Author Comment (AC1) · 15 Apr 2018

Anna Makarewicz

Piotr Kowalczuk

Institute of Oceanology Polish Academy of Science

ul. Powstańców Warszawy 55

PL-81-712, Sopot, Poland

April 15, 2018

Prof. Dr. Oliver Zielinski,

Associate Editor Ocean Science

Head of Marine Sensor Systems

Vice Director Institute for Chemistry and Biology of the Marine Environment (ICBM)

University of Oldenburg, Germany

Cc/

Natascha Töpfer

Copernicus Publications

Editorial Support

Att. The response to review of manuscript by Makarewicz et al., 2017 submitted to Ocean Science and coded OS-2017-100 done by Reviewer#1, document reference OS-2017-100RC1

Dear Prof. Zielinski,

We thank the reviewers for their constructive comments. We have followed their guidance, and rewritten parts of the manuscript to place the work in better context. We have also gone through the text thoroughly to make any edits to the text to improve the flow and any grammatical errors we found that could be corrected. With this exercise we have also responded to all questions raised by Reviewer#1 and introduced necessary correction in revised manuscript.

The detailed comments to the Reviews are given below. After each of Reviewers single comments, our responses start with **Response:** (in bold)

Detailed Response to review by Reviewer #1:

General comments

The review submitted by Reviewer#1 concentrated mostly on methodological issues that we will explain in a point-by-point response below. The Reviewer#1 raised a concern about the main conclusion of our work that phytoplankton was the main source of the protein-like fraction of fluorescent dissolved organic matter, FDOM. We have come to this conclusion based on the simultaneous in situ measurements of protein-like FDOM fluorescence and chlorophyll $a$ fluorescence performed with two different fluorometers integrated in one measurements system. The correlation coefficient R between those two variables was 0.804,

and the linear relationship between them explained 65% of variance ($R^2$ = 0.65). The regression analysis has been conducted on an extensive data set n = 24990 and results were very significant statistically (p<0.0001). The question put forward by Reviewer#1, that there was a very low determination coefficient values in the regression between protein-like FDOM fluorescence and chlorophyll *a* concentration concerned only analysis performed in case when chlorophyll *a* concentration was measured in discrete water samples. We admit that the relationship was weaker in this case, however it was statistically significant. This difference has been explained on Page 21 (lines 473 – 488) and can be attributed to time and space lag between instrumental measurements and water sample collection, reaching in some cases 1.5 hour and up to 3 nautical miles, respectively. This is caused by the "one instrument in water at a time" safety rule applied in almost all research vessels.

In addition, in both 2014 and 2015 we provided quantitative evidence of tight relationship between optical proxy for phytoplankton chlorophyll a concentration: the total absorption coefficient less due to water at 676 (atot-w(676) ) and protein-like FDOM fluorescence (ICH3). The determination coefficient ranged between 0.423 and 0.860 depending on sampled water masses in a given year, see Figure S4 in supplement. We have also shown that there was a very tight coupling between total absorption coefficient less due to water at 676 nm, $a_{tot-w}(676)$ and chlorophyll *a* fluorescence intensity measured in different water masses in 2014 and 2015, Figure S3 in supplement. The determination coefficient between those variables ranged between 0.394 and 0.915 depending on sampled water masses in a given year. In addition to the quantitative analysis we have shown that there was a significant coupling in vertical distribution of protein-like FDOM fluorescence, ICH3 and $a_{tot-w}(676)$ and chlorophyll *a* fluorescence, see Figure 4, E, F, G, page 20.

Findings presented above have been thoroughly discussed in section 4.4 (Pages 31-34) where we have reviewed recent studies providing evidence based on in situ and mesocosm studies that protein-like FDOM fraction has been controlled by phytoplankton dynamics. Recent studies by Chen et al., (2017) (Science of The Total Environment, v:599–600, pp: 355-363) and Reteletti-Brogi et al, 2018 (Science of The Total Environment, 627, 802-811) presented new data linking high abundance of protein-like fluorescence with autochthonous production by phytoplankton and ice algae in the water of Amerasian Basin of Arctic Ocean, sea ice and under ice water in Canadian Archipelago. However, this evidence has been based on limited samples size collected in single surveys. Our data set is unprecedented in volume acquired in repeated surveys in the same area over several consecutive years, and acquired with a custom-made fluorescence sensor (excitation/emission pairs to detect different CDOM pools) simultaneously (the in situ 3 channel WetStar FDOM fluorometer by WetLabs). Therefore we are confident that our conclusion are based on sound arguments.

At the end of his/her review Reviewer#1 has asked questions (Question 13), why the CDOM absorption is not well correlated with DOC concentrations? Well, the weak but statistically significant relationship between $a_{CDOM}(350)$ and DOC concentrations in the North Atlantic is not necessarily unexpected in waters of Atlantic origin. In principle, the $a_{CDOM}(\lambda)$ and DOC is not correlated on oceanic basin scales (Siegel et al., 2002, Nelson and Siegel 2013). The empirical relationships between the $a_{CDOM}(\lambda)$ and DOC reported in literature (e.g.

Massicotte at al., 2017 and references therein) were driven by concentration gradient in DOC and $a_{CDOM}(\lambda)$ found between terrestrial, freshwater, marine and coastal ocean environments. Most of estimated empirical relationships between DOC and $a_{CDOM}(\lambda)$ were derived for coastal regions where there was a local point source of terrestrial DOC (river outlet). Such relationships published for Arctic were usually established for coastal regions in the vicinity of North American and Siberian Rivers (Amon et al., 2012; Matsuoka et al., 2012; 2013; Gonçalves–Araujo et al., 2015; Pavlov et al., 2016; Mann et al., 2016). The environment-specific determination coefficient in relationships between the $a_{CDOM}(\lambda)$ and DOC estimated by Massicotte at al., (2017) was lowest for pelagic ocean and reached 0.44. Our results are in line with statements from Siegel et al., 2002 and Nelson and Siegel, 2013 that the $a_{CDOM}(\lambda)$ and DOC is not correlated in pelagic ocean, as our study region in the Atlantic water inflow regions has very limited input from land runoff and the salinity range we have covered is very limited and high (with occasional freshening due to sea-ice melt rather than terrestrial runoff).

We have tried the Fichot and Benner (2011, 2012) approach to link spectral slope $S_{275-295}$ with carbon specific $a^*_{CDOM}(\lambda)$, but results were unsatisfactory. Rather we have presented similar non-linear relationship between the $S_{300-600}$ with carbon specific $a^*_{CDOM}(\lambda)$, which worked fairly well, and was consistent with non-linear relationship between those parameters presented by Norman et al., (2011) in Antarctica. Again, the Fichot and Benner approach to link spectral slope $S_{275-295}$ with carbon specific $a^*_{CDOM}(\lambda)$ was derived for Gulf of Mexico, where there were contrasting concentrations and compositional CDOM properties exists between the coast near the Mississippi River mouth and central oligotrophic part of Gulf of Mexico (Carder et al., 1999). In our opinion we have presented a thorough discussion comparing our results with existing literature, (Section 4.3).

We have tried to explain most critical points raised by Reviewer#1 in this section. Detailed responses to all individual questions are given below.

**Detailed responses**:

Abstract 1) page 2 line 45 why you concluded that phytoplankton is the main source of protein-like fluorescence based on a r2=0.36?

**Response**: This question has been addressed in General Comments section of our response letter, see above. We have deleted part of this sentence from the abstract: "$_2$"

2) page 3, line 51. how did you arrive to this conclusion? of the Arctic Ocean (Arrigo et al., 2008), which could potentially contribute to increased production of autochthonous (marine) dissolved organic matter (DOM). what about the ice algae? they will disappear and they also contribute to CDOM.

**Response:** We assume that Reviewer refers to line 61 in the Introduction. In fact, this remark does not refer to our results, and the Reviewer#1 likely wanted this sentence to be clarified. Similar remarks have been noted by Reviewer #1 in question 3, therefore we have decided to rewrite this paragraph in the Introduction. We agree with Reviewer #1 that ice algae can be considered as a potential source of autochthonous CDOM/DOM, (e.g. Granskog et al., 2015; Reteletti-Brogi et al, 2018) and this has been addressed in General Comments as well (see above). Above we have given points supporting our conclusion that phytoplankton is a dominant source of CDOM in the Nordic Seas influenced by Atlantic Waters - we have used the term "study area" to specify this (line 51). We would like to underline that we have been conducting field surveys in the ice-free waters, see section 2.1., as our research vessel is not classified as icebreaker. We neither did present any data nor written any conclusions about the sea ice.

We have changed the corresponding paragraph as follow:

The rapid reduction of summer sea ice in the Arctic Ocean in the past decades has various repercussions on the structure and functioning of the Arctic marine system: forcing changes in physics, biogeochemistry and ecology of this complex oceanic system (Meier et al., 2014). One of the most significant consequence of observed rapid Arctic Ocean transition is an increase in the primary productivity of the Arctic Ocean (Arrigo et al., 2008), which could potentially contribute to increased production of autochthonous (marine) dissolved organic matter (DOM) in ice free and under ice waters. The sea ice is also a source of autochthonous CDOM/DOM, (e.g. Granskog et al., 2015; Anderson and Amon, 2015 Reteletti-Brogi et al, 2018). However DOC produced by sympagic algae has limited effect on overall organic carbon mass balance in the Arctic Ocean, as melting of one meter of sea ice would negligibly change DOC concentration in top 50 m of water column, assuming an averaged DOC content in the ice of 100 μMol C, (Anderson and Amon, 2015). Simultaneously, response of terrestrial ecosystems to temperature increase will accelerate permafrost thaw and increase the riverine discharge, resulting in more allochthonous (terrestrial) DOM being released into the Arctic Ocean (Amon, 2004; Stedmon et al., 2011; Anderson and Amon, 2015; Prowse et al., 2015, and references therein). Terrestrial DOM presents a considerable role in the carbon budget of the Arctic Ocean (Findlay et al., 2015; Stein and Macdonald, 2004), especially in coastal waters and continental shelf with large inflow of terrestrial DOM, which constitutes 80% of total organic carbon delivered by Arctic rivers (Stedmon et al., 2011).

Following references have been added to text:

Amon, R.M.W. 2004. The Role of Dissolved Organic Matter for the Organic Carbon Cycle. the Arctic Ocean, [in:] The organic carbon cycle in the Arctic Ocean, Stein, R., and Macdonald, R. W. (Eds.) Springer, Berlin, Heidelberg Chapter 4, 82-99.

Anderson and Amon, 2015. DOM in the Arctic Ocean. [in:] Biogeochemistry of Marine Dissolved Organic Matter, D. A. Hansell, D. A., and Carlson, C. A. (eds), 609–633.

Granskog, M. A., Nomura, D., Müller, S., Krell, A., Toyota, T., & Hattori, H. (2015). Evidence for significant protein-like dissolved organic matter accumulation in Sea of Okhotsk sea ice. *Annals of Glaciology*, *56*(69), 1–8. https://doi.org/10.3189/2015AoG69A002

Retelletti-Brogi, S., S-Y. Ha, K. Kim, M. Derrien, Y.K. Lee, and J. Hur, 2018. Optical and molecular characterization of dissolved organic matter (DOM) in the Arctic ice core and the

underlying seawater (Cambridge Bay, Canada): Implication for increased autochthonous DOM during ice melting. Science of the Total Environment 627, 802–811

3) page 3 line 65, which percentage to carbon budget? DOM presents a considerable role in the carbon budget of...

**Response**: We agree with Reviewer#1 that this sentence wasn't clear. However, it is beyond the scope of this paper to place exact numbers on budget terms that have large margins of error even in the most up-to-date estimate of the carbon budget of the Arctic Ocean. We have changed it as follow:
Terrestrial DOM presents a considerable role in the carbon budget of the Arctic Ocean (Findlay et al., 2015; Stein and Macdonald, 2004), especially in coastal waters and continental shelf with large inflow of terrestrial DOM, which constitutes 80% of total organic carbon delivered by Arctic rivers (Stedmon et al., 2011).

4) line 71. please add Pegau reference to this list Hill, 2008; Granskog et al., 2007,

**Response**: Agree. The reference to:
Pegau,W. S. (2002), Inherent optical properties of the central Arctic surface waters, J.Geophys. Res., 107(C10), 8035, doi:10.1029/2000JC000382.
has been added to the revised manuscript text and reference list.

5) line 73 sorry this is not conclusive. UV can also produce radicals after interacting with CDOM resulting in more toxic and damaging effects! and preserves marine ecosystem from harmful ultraviolet radiation

**Response**: We agree with Reviewer suggestion. The sentence has been rewritten as follows:

Particularly in absence of sea ice, light absorbed by CDOM in visible part of the spectrum limits the light available for photosynthetic organisms (Arrigo and Brown, 1996), but also shields marine ecosystem from potentially harmful ultraviolet radiation strongly absorbing electromagnetic radiation in UVB and UVA (Erickson III et al., 2015). CDOM is also important substrate in photochemical reactions contributing to direct remineralization of organic carbon, production of bioavailable low molecular weight DOM but also formation of reactive oxygen species that could potentially be toxic to marine organisms (Mopper and Kieber, 2002, Kieber et al., 2003, Zepp, 2003).

Following references have been added to references list:

Arrigo K. and C. Brown, 1996. Impact of chromophoric dissolved organic matter on UV inhibition of primary productivity in the sea. Marine Ecology Progress Series, 140, 207-2016

Kieber, D.J., Peake, B.M., Scully, N.M., 2003. Reactive oxygen species in aquatic ecosystems. In: Helbling, E.W., Zagarese, H. (Eds.), UV Effects in Aquatic Organisms. Royal Society of Chemistry, Cambridge, pp. 251– 288.

Mopper, K., Kieber, D.J., 2002. Photochemistry and the cycling of carbon, sulfur, nitrogen and phosphorus. In: Hansell, D.A., Carlson, C.A. (Eds.), Biogeochemistry of Marine Dissolved Organic Matter. Academic Press, New York, pp. 455– 507.

Zepp, R.G., 2003. Solar ultraviolet radiation and aquatic biogeochemical cycles. In: Helbling, E.W., Zagarese, H. (Eds.), UV Effects in Aquatic Organisms and Ecosystems, vol. 1. The Royal Society of Chemistry, Cambridge UK, pp. 137– 184.

6 )line 78, what fraction of DOM is CDOM? what fraction of CDOM is FCDOM?

**Response**: This is a good question, to which there is no consensus answer within the community working with DOM, and providing detailed answer to this question is beyond the scope of this paper. Stedmon and Nelson (2015) in their most recent book chapter presented only a qualitative schematic drawing of dissolved organic matter with subdivision for its chromophoric and fluorescent part with indication of elemental carbon, nitrogen, phosphorus, hydrogen and sulphur contribution. Unfortunately, no quantitative information is given (neither likely available). Similarly, the FDOM fraction has not been quantified as percent of CDOM or DOM (Stedmon and Nelson, 2015). Nelson and Siegel (2013) have defined CDOM as:

"Chromophoric dissolved organic matter (CDOM; also often referred to as gelbstoff or gilvin) is the fraction of DOM that interacts with solar radiation. CDOM compounds absorb light, and a fraction of them are also fluorescent. For the purposes of this review, we operationally define CDOM as material that passes through a submicron filter (usually 0.2–0.4 μm) and appreciably absorbs light in the solar radiation bands as found at the Earth's surface (e.g., UVB, UVA, and visible light; 280–700 nm). This definition practically excludes much of the DOM pool, which spectroscopically absorbs shortwave UV radiation but does not interact with light in the natural environment (Fichot & Benner 2011). We further operationally define the quantity of CDOM by its Naperian absorption coefficient at a reference wavelength. Quantification of CDOM in terms of mass or carbon content is not currently possible, so obviously optical characterization of any nature is relative to the exact composition of CDOM, which likely varies in both time and space."

Our definition, given in the Introduction comprise in a shorter form of definition given by Nelson and Siegel (2013). The research community has also consistently used optical properties to characterize CDOM in oceanic environments, following concepts developed by Jerlov (1968) over 50 years ago.

7) page 4 line 100, upstream you meant? changes associated with CDOM in the areas downstream of the Atlantic Water inflow region

**Response:** We agree with Reviewer#1. North Atlantic south of Nordic Seas, are upstream in the North Atlantic Current. Changed accordingly.

8) page 8 line 209, S between 300 and 600 nm line 218 why additional acdom375 and acdom443 when actually the range is 300 and 600 nm? line 224 why to use micron$^{-1}$ use nm$^{-1}$

**Response:** We have calculated spectral slope coefficient in the spectral range 300 – 600 nm. We have included additional CDOM absorption coefficient values at 375 and 443 nm, $a_{CDOM}(375)$ and $a_{CDOM}(443)$, to enable direct comparison of results with presented in other relevant studies e.g. Stedmon and Markager, 2001, Matsuoka et al., 2011 2012, 2013, 2017; Granskog et al., 2012; Hancke et al., 2014; Gonçalves–Araujo et al., 2015; Pavlov et al., 2015. The values of the slope coefficient have been scaled by multiplying by 1000, and given

in units μm-1 for better visualization in tables and figures, which is consistent with Stedmon and Markager, 2001, 2003, Kowalczuk et al., 2006, Stedmon and Nelson, 2015 among others.

9) page 9 equation 3, why the use of spectrophometry for chl? this is an old technique that has a larger error and is less sensitive than fluorometry or HPLC. What is the error of this emthod? Did you compare this method with HPLC or fluorometry nonacidification technique?

**Response:** We agree with Reviewer#1 that the HPLC method is most accurate way for estimation of chlorophyll $a$ concentration, however due to its high cost and time-consuming analysis, we have chosen the spectrophotometric method, because it is simple, fast, low-cost, and not dependent on external standards. The spectrophotometric method was the most convenient way for processing large number of collected samples. We worked in mesotrophic and eutrophic waters, where the chlorophyll $a$ concentration varied between 0.1 to 15 mg m$^{-3}$. The spectrophotometric method for determination of chlorophyll $a$ concentration originally developed by Lorentzen (1967) has been recommended for use in mesotrophic and eutrophic waters by "Guidelines for the Baltic monitoring program". Baltic Sea Environment Proceedings, 27D, Helsinki Commission, Publication BSEP27D, Helsinki, 1988. We agree that fluorometric method of chlorophyll $a$ concentration is more sensitive, however it is also heavily dependent on the fluorescence quantum yield, that is different for various phytoplankton groups and the fluorometer must be routinely calibrated against the external standards. Recently, there has been observed a rapid and dramatic change in phytoplankton phenology in Nordic Seas, where dominant diatoms have been replaced by coccolithophores advancing northward with warm Atlantic Water (Oziel et al., 2017). So, in our opinion the fluorometric method of chlorophyll $a$ concentration measurements could be biased by variable quantum yields. The spectrophotometric method based on extracted pigments absorption measurements is also much closer to measured optical Chl a proxies e.g. $a_{tot-w}(676)$.

The comparison between spectrophotometric method and HPLC method of chlorophyll $a$ concentration measurements used in our lab has been previously presented by Darecki and Stramski, 2004, who found very good agreement between those two methods.

10) page 9 equation 4, I disagree. You cannot mix apples with bananas. DOC is not DOM unless you estimate DOM based on DOC with a curve or factor., same for equation 5

**Response**: We disagree. Specific DOC absorption (a* or SUVA) is a standard way adopted to examine CDOM properties in many studies measuring CDOM and DOC, and can provide insights about the quality of CDOM relative to DOC (Weishaar et al., 2003; Stedmon and Nelson , 2015; Massicotte et al., 2017). Principle goal of our study was to characterize the CDOM and FDOM optical properties and to identify their primary sources. As we mentioned in the answer to question 6, contributions of CDOM and FDOM to DOM (or DOC are generally unknown. Both CDOM and FDOM are characterized in aquatic environment through optical properties, Siegel and Nelson (2013). Optical properties normalized to DOC concentration are so called specific (in our case specific to carbon) optical properties and express the absorption crossection of the unit of mass of the substance. This specific optical crossection could be used as a measure of converting optical biogeochemical proxy into a concentration of substance (in this case carbon). Some of the specific optical properties have also biogeochemical meaning because they are related to diagenetic state of the substance, its chemical composition and molecular weight. Therefore, these variables were included in our manuscript. Stedmon and Nelson (2015) as well as Massicotte et al. (2017) advocated for

a use of those variables and ancillary parameters useful to characterize CDOM/FDOM and helpful in source identification. Therefore, we included those variables in our manuscript as they are relevant for the CDOM community, and would like to keep them.

11) line 305, how did you calculate the offset of wetstar-3 fluorescence measurements? reference with respect to nanopure? constant temperature?

**Response:** We have estimated a time drift of the fluorometer response, which was calculated by difference in raw counts values measured within similar pressure, salinity and temperature ranges (at ca. 200 m depth, temperature, 6.5 deg C. salinity >34.9) in the core of Atlantic Water, which we assume has not changed between years. The average difference in measured raw counts values in each channel in 2015 relative to 2014 was attributed to a drift of the optical detector, causing a deterioration of sensitivity and increase of raw counts. We calculated the average difference, and this was subtracted from all recorded raw counts in each channel measured during 2015 survey.

12) S slope without units?

**Response**: We could not identify the manuscript line where spectral slope unit was missing.

13) figure 3 is hard to interpret due to the vertical variability of properties

**Response**: We have foreseen this problem and Figure 3a presented a vertical distribution of sea water properties, giving a color scale of depth as third variable. An example of vertical distributions of ICH1 is shown in Figure 4.

14) many questions but fewer explanations or explanation attempts. beyond sampling aliasing, Why not acdom350 not well related to DOC? why not links between s275-295 and DOC? what is the linkage between particulate iron and absorption slopes?

**Response:** We have responded to this question in more detail in the General Response section (See above). In brief, we sampled a pelagic environment with narrow salinity range, and we believe that this in part explains why we cannot use CDOM as predictor of DOC. The area of study is not substantially influenced by riverine sources., therefore the we did expect that CDOM optical properties will predicts DOC in this environment with high accuracy.

At this point we can also address the last question concerning the iron. First of all, we have been analyzing properties of dissolved substances not particulate, therefore particulate iron did not affect our CDOM absorption measurements. At this point we could expresses how much useful was inclusion of SUVA254 variable in our analysis. Stedmon and Nelson (2015) in their book chapter has stated the variability of SUVA254 is between $0.5 - 5$ m$^2$ g$^{-1}$C in oceanic environment, and values over the 5 m$^2$ g$^{-1}$C indicated the possible interference of dissolved iron on optical properties of CDOM. In this study SUVA254 was in the range of $0.56 - 2.54$ m$^2$ g$^{-1}$C, with average value of ca. 1.7 m$^2$ g$^{-1}$C, which is a typical value for pelagic ocean (Massicotte et al., 2017). Based on this we can conclude that iron in dissolved and particulate phase had a negligible effect on CDOM/FDOM optical properties in our study area.

---

## Author Comment (AC2) · 16 Apr 2018

Anna Makarewicz

Piotr Kowalczuk

Institute of Oceanology Polish Academy of Science

ul. Powstańców Warszawy 55

PL-81-712, Sopot, Poland

April 15, 2018

Prof. Dr. Oliver Zielinski,

Associate Editor Ocean Science

Head of Marine Sensor Systems

Vice Director Institute for Chemistry and Biology of the Marine Environment (ICBM)

University of Oldenburg, Germany

Cc/

Natascha Töpfer

Copernicus Publications

Editorial Support

Att. The response to review of manuscript by Makarewicz et al., 2017 submitted to Ocean Science and coded OS-2017-100 done by Reviewer#2, document reference OS-2017-100RC2

Dear Prof. Zielinski,

We thank the reviewers for their constructive comments. We have followed their guidance, and rewritten parts of the manuscript to place the work in better context. We have also gone through the text thoroughly to make any edits to the text to improve the flow and any grammatical errors we found that could be corrected. With this exercise we have also responded to all questions raised by Reviewer#2 and introduced necessary corrections in the revised manuscript.

Response to general comments by Reviewer#2

The main critical point raised by Reviewer#2 was that we have failed to prove that observed interannual variability of CDOM optical properties was driven by large scale oceanic circulation. We have revised results section and we have added statistical analysis of variance requested by Reviewer#2 to highlight the statistical significance of observed interannual variability. We think that this information supports our statement. We know that definite proof could only be given if we knew the water masses history. Nordic Seas are a very complex and dynamic region and such analysis is very complex and beyond scope of this study. It should be underlined that numerous water masses that interface in Nordic Seas are in constant motion and their properties undergo continuous transformations due to thermodynamic phenomena in the polar zone (loss of heat content, freezing, and melting of ice cover) and dynamic phenomena driven by thermodynamics (loss of buoyancy and deep thermohaline convective mixing). Such complex physical processes and fluctuations of their intensity make interpretation of measurement of other physical parameters of oceanic waters

very difficult. Each of the identified water masses in the study area have had its history, which is beyond the scope of this study to analyze in detail. We were only capable to present a snapshot of current state of optical properties during the field campaigns. The main factor influencing the whole environment of Nordic Seas is an inflow of warm Atlantic waters. We have shown a co-incidence that the decrease of CDOM absorption occurred during intense inflow of Atlantic waters. We do admit that based on our data set we could not give a definitive proof of e.g. correlation with temperature or salinity anomalies, simply because the optical measurements time series were to scarce compared to number of hydrographic observations. Furthermore, temperature is not a conservative tracer of water masses.

Modification of DOM optical properties in a given water mass is superimposed on large scale circulation and dynamics. It is very important to underline that processes of in situ production, transformation and decomposition of DOM occur at different time scales and usually are delayed in phase (Nelson et al., 1998, Jorgensen et al, 2014). We have shown that a DOM fraction – protein-like substances - are produced by phytoplankton given the very strong correlation between fluorescence intensity of this fraction and chlorophyll $a$ concentration. Simultaneously we did not observed any significant correlation between CDOM absorption at 350, $a_{CDOM}(350)$ with chlorophyll $a$ concentration. This paradox can be explained by chemical properties of two different CDOM fractions, as FDOM is a sub-fraction of CDOM. The fluorescent amino acids have their excitation (absorption) band in UV-B between 260-275 nm, and belong to labile DOM. These compounds usually have a low number of aromatic rings built in their chemical structures. The absorption band of chromophoric dissolved organic compounds shift toward longer wavelengths and broadens its spectral width with increasing number of aromatic rings and increased number of conjugated bonds in delocalized molecular orbitals (Woźniak and Dera, 2007). Microbial transformation of DOM leads to creation of more condensed and more aromatic ("humic") DOM, that is characterized with absorption bands at longer wavelengths (in UV-C and visible part of the spectrum). The microbial transformation of labile and semi-labile DOM occurs on time scales ranging from three months (Nelson et al., 1998) to over a year (Jorgensen et al., 2014). During this time, in the Nordic Sea the given water mass where protein-like FDOM was produced have fair chance, to be transported further north to the Central Arctic Ocean basins, or be submerged into abyss during convective mixing events during winter. Therefore, likely due to temporal mismatch between in situ DOM production and its further transformation we could not observe correlation between, $a_{CDOM}(350)$ with chlorophyll $a$ concentration. In our case $a_{CDOM}(350)$ represented past memory of DOM production and transformation process that occurred with variable intensity upstream in the North Atlantic Current, and the fluorescence intensity of protein-like DOM fraction represented a proxy of initial production processes. This interpretation does not contradict with the statement that phytoplankton growth is the main source of FDOM/CDOM in Nordic Seas.

The detailed comments to the Review#2 are given below. After each of Reviewer#2 single comments, our responses start with **Response:** (in bold)

**Detailed response**:

L. 83: what do the authors mean by efficient? Please, be more specific, otherwise the reader can read interpret it as the other methods are not efficient.

**Response:** We agree with Reviewer suggestion. The sentence has been rewritten as follows: "Use of in situ DOM fluorometers enables  low cost and high sample rate observations of distribution of FDOM and related biogeochemical proxies with greater temporal and spatial resolution (Belzile et al., 2006; Kowalczuk et al., 2010)"

L. 93-94: the contrasting optically properties of AW and PW are also with respect to the FDOM fraction (see Jørgensen et al 2014; Gonçalves-Araujo et al 2016). Additionally, such a contrast in CDOM properties is also highlighted in Stedmon et al 2015.

**Response:** We appreciate Reviewer suggestion and references have been added as follows: "Optically these waters are contrasting especially with respect to CDOM (Granskog et al., 2012; Pavlov et al., 2015, Stedmon et al., 2015) and FDOM (Jørgensen et al., 2014; Gonçalves-Araujo et al., 2016)."

The reference to has been added to the revised manuscript text and reference list:
Jørgensen, L., Stedmon, C. A., Granskog, M. A. & Middelboe, M. Tracing the long-term microbial production of recalcitrant fluorescent dissolved organic matter in seawater. Geophys. Res. Lett. 41, 2481–2488, 2014.
Stedmon C. A., Granskog, M.A., and Dodd, P. A.: An approach to estimate the freshwater contribution from glacial melt and precipitation in East Greenland shelf waters using colored dissolved organic matter (CDOM). J. Geophys. Res.: Oceans, 120 (2), 1107-1117, doi.org/10.1002/2014JC010501, 2015b

L. 94-97: missing references

**Response:** We have added the following three references:

Skogen, M.D., Budgell, W.P. and Rey, F. Interannual variability in Nordic seas primary production. ICES Journal of Marine Science, 64(5), 889-898, 2007.

Olsen, E., Aanes, S., Mehl, S., Holst, J.C., Aglen, A. and Gjøsæter, H. Cod, haddock, saithe, herring, and capelin in the Barents Sea and adjacent waters: a review of the biological value of the area. ICES Journal of Marine Science, 67(1), pp.87-101, 2009.

Dalpadado, P., Arrigo, K.R., Hjøllo, S.S., Rey, F., Ingvaldsen, R.B., Sperfeld, E., van Dijken, G.L., Stige, L.C., Olsen, A. and Ottersen, G. Productivity in the Barents Sea-response to recent climate variability. PloS one, 9(5), p.e95273, 2014.

L. 100: would be interesting to shown how CDOM/FDOM measurements can be used as a proxy for tracing other parameters (e.g., freshwater, DOC, phytoplankton primary production, etc.).

**Response:** We agree with reviewer. This sentence has been revised as follows:.

"In context of ongoing and further anticipated intensification of Atlantic Ocean inflow to the Arctic Ocean, description of processes and factors controlling CDOM/FDOM properties and distribution could be used to better predict future changes associated with CDOM in the areas

downstream of the Atlantic Water inflow region inflow region, estimation of glacial melt water (Stedmon et. al.,2015) and tracing water masses (Gonçalves-Araujo et al., 2016)."

L. 102-118: There is a recent paper that may be useful for the authors: Gonçalves-Araujo et al 2018.

**Response:** According to the reviewer's suggestion, recent paper of Gonçalves-Araujo et al., 2018, has been included in the discussion section.

L. 110-113: please consider re-writing the sentence. It is missing a verb.

**Response:** Missing verb has been added to the sentence.
"Seasonal studies on CDOM contribution to overall variability of inherent optical properties (IOPs) were reported in sea ice (Kowalczuk et al., 2017) and in the water column during a spring under–ice phytoplankton bloom north of Svalbard (Pavlov et al., 2017)."

L. 139: consider adding the (Bjørnøya Current) BC to figure 1.

**Response:** Bjørnøya Current (BC) has been added to figure 1.

[Figure]

Figure 1: Map of the sampling stations during AREX2013 (blue circles), AREX2014 (green circles), AREX2015 (red circles) with general surface circulation patterns in the Nordic Seas. Atlantic Waters: WSC, West Spitsbergen Current. Polar waters: ESC, East Spitsbergen Current; SC, Sørkapp Current; EGC, East Greenland Current; BC, Bjørnøya Current; YP, Yermak Plateau; SF, Storfjorden; SPB, Spitsbergenbanken.

L. 171-172: the water sample collected below the Chl-a maximum. Authors can provide more information regarding it. Was it within the pycnocline? Or at the bottom of the deep Chla maximum layer? How was the sampling depth determined?

**Response:** The information concerning the range of depth of chlorophyll *a* samples has been described in detail in lines 169-173 and we feel this provides sufficient details:

"Samples were collected at three depths: near the surface, ca. 2 m depth, at chlorophyll *a* maximum, that was usually located between 15 and 25 m depth, and below chlorophyll *a*

maximum, between 50 and 70 m. The exact position of chlorophyll *a* maximum depth was estimated from vertical profile of chlorophyll *a* fluorescence during CTD downcast."

The sampling depth below the chlorophyll *a* maximum depth was determined visually based on the vertical profile of chlorophyll *a* fluorescence during CTD downcast. The vertical salinity profiles in core of AW water do not have detectable pycnocline within top 200 m, see Figure 4. The pycnocline associated with melt water overlying AW was observed in the frontal zone within marginal ice zone, Figure 4. The phytoplankton bloom associated with the marginal ice zone has its subsurface maximum within the pycnocline at 15-25 m, as shown on vertical profiles of chlorophyll *a* fluorescence, Figure 4. The subsurface chlorophyll *a* maximum in the marginal ice zone and core of AW waters was located in a similar depth range.

L. 178-189: have the authors performed any test to check whether the differences in storage affected their results?

**Response:** We did not perform any test to check how storage can influence the results. However, according to Stedmon and Markager, 2001 a few months of storage has little or no effect on CDOM absorption spectrum compared to the possibility of spectra disturbances onboard due to vibration, pitch and roll of the ship. Given the delays were similar between cruises the data are still comparable. We also prefer this over freezing samples, which modify the optical properties of DOM even more.

L. 250: was is the meaning of CRM?

**Response:** CRM is the abbreviation for consensus reference material. CRM is used as quality control of DOC measurements internationally. The explanation of the abbreviation has been added to the manuscript as follows:

"Consensus reference material (CRM) supplied by Hansell Laboratory from University of Miami was performed as quality control of DOC concentrations. The methodology provided sufficient accuracy(average recovery 95%; n = 5; CRM = 44 - 46 μM C; our results = 42 - 43 μM C) and precision represented by a relative standard deviation (RSD) of 2%."

L. 303: Sagan et al 2017. Authors should avoid citing unpublished work.

**Response:** We agree with Reviewer suggestion. Citation has been changed to: "(Sagan S., *personal communication, 2017*)".

L. 315-317: It is worth to mention that the referred water masses are found within the "upper layer", given that no water mass from deep layers was presented.

**Response:** The sentence has been rewritten as follows: "The epipelagic layer of the Nordic Seas is dominated by AW and PSW, and waters formed in the mixing process and local modifications (precipitation, sea–ice melt, riverine run–off, and surface heating or cooling) of these two water masses."

L. 324: the meaning of the PSWw (as well as its thermohaline characteristic intervals) was not presented. Additionally, and for consistency the PSWw is mostly referred in the literature as ASW (e.g. Pavlov et al 2015 and Gonçalves-Araujo et al 2016).

**Response:** We appreciate Reviewer suggestion and explanation of the PSWw abbreviation has been added to the revised manuscript as follows:

"Part of AW (except Polar Surface Water warm, PSWw) included waters with density below σθ=27.7 kgm-3 (marked on Figure 3 with dashed isopycnal line) used by Rudels et al (2005) as a threshold value between AW and PW."

Thermohaline characteristics for all water masses defined by Rudels et al. (2005) were presented in Table S1 in Supplement. Additionally detailed description and thermohaline characteristics of PSWw were presented in lines 330-334. Furthermore, a note about ASW has been added to the revised manuscript:

"Warmer PSWw has been considered here with the same σθ≤27.7 kg m-3 criterion and Θ>0°C (Rudels et al., 2005), due to summer season measurements and higher temperatures of low salinity surface waters in the eastern Fram Strait. Furthermore PWSw was also limited to the uppermost 50 m of the water column with S≤34.9. The water mass with similar TS characteristics to PSWw but slightly different limits was referred in the literature as Arctic Surface Water, ASW (e.g. Pavlov et al 2015 and Gonçalves-Araujo et al 2016) but due to the dominance in the area of water originating from Atlantic Ocean the name PSWw from Rudels et al., (2005) classification is used."

L. 339: I suggest the authors to add some specific words to the title that would better reflect the results presented in this sub-section: spatial variability and hydrography.

**Response:** Thank you. According to Reviewer suggestion and the title of sub-section 3.1. has been changed as follows:

"*3.1. Interannual and spatial variability of CDOM absorption in Nordic Seas surface waters in relation to hydrography.*"

L. 340-357: please consider including the standard deviation or standard error to the averaged values presented through the text.

**Response:** Done accordingly. The required information has been added in the revised manuscript.

L. 353: 200 m water layer. The indication for meter is missing.

**Response:** The unit has been added to the revised manuscript: "(…) and salinity time series in the top 200 m water layer (Walczowski et al., 2017)." Thank you.

L. 358: what do the authors mean by distinct? Please be more specific.

**Response:** We agree with Reviewer suggestion that the word "distinct" has been used in a non-clear context. The sentence has been rewritten as follows:

„In 2015, SC and ESC branches originating from the Barents Sea were pronounced, as indicated by lower temperature and salinity, Figure 2c, were distinct leading to elevated $a_{CDOM}(350)$ values on the West Spitsbergen Shelf and along the section from Sørkapp down to 74°N and near Bjørnøya Island."

L. 371-372: How do the authors explain the higher CDOM in PSWw compared to PSW? Shouldn't the DOM signal in PSWw be diluted by sea-ice melt waters and therefore, lower

than the DOM observed in the core of PSW as highlighted in Pavlov et al 2015, Stedmon et al 2015, Granskog et al 2012, Gonçalves-Araujo et al 2016, etc.?

**Response:** According to Rudels classification PSWw originates from AW water during its cooling during winter and mixing with PW in the Barents Sea and Nansen Basin and distinguished from PSW by higher temperature in the similar salinity range, see Figure 3. Pavlov et al., (2017) and Kowalczuk et al., (2017) and also Gonçalves-Araujo et al 2018, have provided data that PSWw north of Svalbard and in Nansen Basin do not differ very much from AW in terms of inherent optical properties. We agree that in general, CDOM absorption in PSWw should be lower compared to CDOM absorption in PSW, however in our study PSW was represented by very low samples number and those variations between PSW and PSWw were not statistically significant in 2013 (p=0.631005, T-Test).

L. 378-379: highest aCDOM(350) found within PSW and AAW in 2015...are those values for each water mass significantly similar? How do the authors explain such a high CDOM content associated to the AAW?

**Response:** In 2015, most of averaged values of $a_{CDOM}(350)$ between classified water masses show significant differences (Table r1). Only PSW and AAW are not significantly different (p>0.05) and shows high similarity between each other. This may be due to the low number of samples in a given class. Additionally, AAW in 2015 has a wide range of depths between 5-257 m with mean depth 76±76m.

Table r1: P-values obtained as a result of the T-test between classified water masses in 2015. Red numbers mean p>0.05.

| Year | Water masses | t-value | df | p | significance |
|---|---|---|---|---|---|
| 2015 | AW vs. PSW | -4.44626 | 160 | 0.000016 | S |
| | AW vs. PSWw | -4.06548 | 227 | 0.000066 | S |
| | AW vs. AAW | -4.78185 | 163 | 0.000004 | S |
| | AW vs. IW/DW | 3.248989 | 173 | 0.001392 | S |
| | PSW vs. PSWw | 2.620125 | 77 | 0.010584 | S |
| | PSW vs. AAW | 0.295868 | 13 | 0.772004 | NS |
| | PSW vs. IW/DW | 4.306062 | 23 | 0.000263 | S |
| | PSWw vs. AAW | -2.64463 | 80 | 0.009840 | S |
| | PSWw vs. IW/DW | 4.851875 | 90 | 0.000005 | S |
| | AAW vs. IW/DW | 5.090324 | 26 | 0.000026 | S |

L. 380-382: Could this represent a consistency in the DOM content (fingerprint) with respect to each water mass?

**Response:** The similarity of the variability ranges of the optical characteristics for 2013 and 2015 is not consistent within each water mass, except in PSW (p=0.806317) based on T-test. Variability range in 2013 is much wider towards higher values than in 2015. Each year represented a different optical and hydrographic situation. The year 2013 and 2015 are similar through higher CDOM values in comparison to 2014 and the visible effect of low salinity water from SC current on the West Spitsbergen Shelf. This sentence may confuse readers and therefore have been removed from the manuscript:

"There were similarities between 2013 and 2015 regarding variability ranges of CDOM optical characteristics in the water masses and distribution of low saline waters from ESC and SC."

L. 386-391: Since SUVA254 and a350* are both normalized by the same value (DOC concentration), what do the differences between the two parameters mean?

**Response:** Both spectral carbon specific absorption coefficients were normalized to the same DOC concentration values. They have the same physical meaning, and represent absorption cross section of a mass unit of element at different wavelengths. However those specific absorption coefficients have different meaning from biogeochemical point of view. The $SUVA_{254}$ is related to saturation of DOM mixture with aromatic rings (Weishaar et al 2003), and $a*_{CDOM}(350)$ has been frequently used for remote sensing application for DOC determination with use satellite imagery (Fichot and Benner, 2011). This paragraph was not intended to reveal the differences between $a*_{CDOM}(350)$ and $SUVA_{254}$ but to report the existing state of various optical parameters in the Nordic Seas and to indicate differences or similarities between years. Both of the parameters ($a*_{CDOM}(350)$ and $SUVA_{254}$) were useful in discussion on CDOM origin in section 4.3 and 4.2, respectively.

L. 396-397: is that decrease significant?

**Response:** The interannual changes in DOC describe a significant decrease between 2013 and 2015 (p<0.000001, T-Test) and 2014 and 2015 (p<0.000001, T-Test). The difference in DOC concentration between 2013 and 2014 is not significant (p=0.314, T-Test). The sentence has been modified as follows:
"The average DOC concentration in the study area was highest in 2013 (80.69 μmol/L) and decreased significantly (p<0.000001) year by year (Table 3) to 67.64 μmol/L in 2015."

L. 396-399: what was the correlation between a350 and Chla? If autochthonous DOM is dominant in the sampling region, there should be a good correlation between the two parameters, right? (or at least for AW, with little continental/Arctic influence)

**Response:** There are studies that reported the correlation between $a_{CDOM}(\lambda)$ and chlorophyll *a* concentrations e.g. Meler et al., 2017; Dall'Olmo et al., 2017, but in the current study we did not observe correlation between $a_{CDOM}(350)$ and chlorophyll *a* concentration in the Nordic Sea during study period. In the General Response we have explained why such a statistical relationship has not been observed in the Nordic Seas. Simply the time scales of water transport mixing, convection and water properties transformations due to loss of heat content are shorter than time scales required for transformation of labile and semi-labile DOM into more condensed and aromatic refractory DOM by bacteria.

Fig 3b: It is clear that the Ich1 had the highest fluorescence intensity values in R.C., but, since the results of this MS are more focused on the Ich3 (autochthonous), I suggest the authors to present the Ich3 in this figure as well.

**Response:** Thank you. The figure for $I_{CH3}$ has been transferred from the supplement to the manuscript together with description:

"In case of protein-like FDOM the highest values were observed in PSW, PSWw mid depth (15-50m,what can be associated with chlorophyll a maximum) and in part of AW, which was

separated from PSWw (upper part: T>0, σθ≤27.7, S>34.9). The lowest protein-like FDOM values were observed in AW (lower part: 27.7<σθ≤27.97) and in PSWw where σθ≤26.5 (Figure S2b)."

L. 469-470: "The relationship was more significant in 2014...". It looks like that the significance is the same for both years. Did the authors want to say that the relationship was stronger (given differences in r2)?

**Response:** We agree with Reviewer and the sentence has been rewritten as follows:
"The relationship was  stronger in 2014 ($R^2$ = 0.75, p<0.0001, n = 17700, blue line in Figure 5a, Table 4) when broader  influence of AW water was observed (Walczowski et al., 2017), than in 2015 ($R^2$ = 0.45, p<0.0001, n = 7290, red line in Figure 5a)."

Fig 5a: Looking at the legend it is not clear what the black line means. Is it the regression curve for the combined dataset from 2014 and 2015 or was it the difference between them? I also think that the information displayed in Table 4 can be easily incorporated to Fig 5.

**Response:** According to the Reviewer's advice the legend in Figure 5 has been modified. The Black line corresponds to a regression line for combined datasets from 2014 and 2015. The information displayed in Table 4 has been combined with the Figure 5.

L. 508: At what salinity did Grankog et al 2007 found acdom(355) greater than 15m-1? It would make it easier to establish comparison to the other studies.

**Response:** The required information about salinity has been added in the revised manuscript as follows:

"Exceptionally high CDOM absorption has been also observed in the southern part of Hudson Bay near river outlets with $a$CDOM(355)> 15 m$^{-1}$, at salinity close to 0 (Granskog et al., 2007)."

L. 502-518: There is a recent paper from Gonçalves-Araujo et al 2018 that shows aCDOM values for the central Arctic and surroundings that the authors may want to look at.

**Response:** We have used the information presented in the paper by Gonçalves-Araujo et al 2018 in the Discussion section. We have updated the discussion to include this as follows:

The highest CDOM absorption in the Arctic Ocean has been observed in coastal margins along Siberian Shelf in Laptev Seas, close to Lena River delta; $a_{CDOM}$(440) = 2.97 m$^{-1}$, salinity close to 0, (Gonçalves–Araujo et al., 2015) and in Laptev Sea Shelf Water at the surface; $a_{CDOM}$ (443)>1m$^{-1}$, salinity <28, (Gonçalves–Araujo et al., 2018) and at the coast of Chukchi Sea and Southern Beaufort Sea influenced by riverine inputs of Yukon and Mackenzie Rivers; aCDOM(440) > 1 m$^{-1}$, salinity <28, (Matsuoka et al., 2011, 2012; Bélanger et al., 2013). Exceptionally high CDOM absorption has been also observed in the coastal Hudson Bay near rivers outlets with $a_{CDOM}$(355)> 15 m$^{-1}$, salinity close to 0 (Granskog et al., 2007). Pavlov et al. (2016) reported $a_{CDOM}$(350) of up to 10 m$^{-1}$ at salinity of 21 in surface waters of the White Sea. Terrestrial CDOM from Siberian Shelf has been diluted and $a_{CDOM}$(440) decreased to ca. 0.12 m$^{-1}$, at salinities 32.6 (Gonçalves–Araujo et al., 2015) and transported further toward the Fram Strait by the Transpolar Drift being gradually diluted or removed (Stedmon et al., 2011;

Granskog et al., 2012). In the Transpolar Drift and the central AO, CDOM absorption in surface waters was dominated by terrestrial sources with observed $a_{CDOM}$(443) values varied between ~0.15 m $^{-1}$, at salinities close to +/-27 (Lund–Hansen et al., 2015) and ~0.5 m$^{-1}$ at salinity range from 26.5 to 29.5 (Gonçalves-Araujo et al., 2018). Dilution also effectively decreased CDOM absorption in western Arctic Ocean, and average CDOM absorption in the Chukchi Sea and Beaufort Seas was $a_{CDOM}$(440) = 0.046 m$^{-1}$, at salinities > 32.3 (Matsuoka et al., 2011, 2012; Bélanger et al., 2013).

Also we have added the sentence at the end of mentioned paragraph:

"The influence of transformed Atlantic Water generated in the Barents and Norwegian Sea had impacted on $a_{CDOM}$(443) values in the Beaufort Gyre and Amundsen and Nansen basins, causing its decrease below 0.2 m$^{-1}$ as reported by Gonçalves–Araujo et al., (2018).

L. 521: do the values also agree with results published in Stedmon et al 2015?

**Response:** The $a_{CDOM}$(350) values range (~0.1- 0.2 m$^{-1}$) from the western part of Fram Strait section reported by Stedmon et al. 2015 also agree with our results from AREX2014. The citation has been added to the revised manuscript.

"The reported lower range of $a_{CDOM}$(350) observed in AW during AREX2014 (2014: 0.14±0.06 m $^{-1}$) is in good agreement with data from eastern part of Fram Strait at 79°N section reported by Granskog et al. (2012), Stedmon et al., (2015) and Pavlov et al. (2015) and with data reported by Hancke et al. (2014) south of the Polar Front in the Barents Sea."

L. 541: "...and similar features of CDOM properties were..." - What do you mean by similar features of CDOM properties?

**Response:** We have used wrong phrase for description of similar variability ranges of CDOM absorption coefficient and spectral slope coefficient. Pavlov at el. (2017) mentioned the lack of differences in optical properties (absorption coefficient values) for water with lower salinity and lower temperature and Atlantic water. A similar relationship was observed in our study. This sentence has been rewritten as follows:

"Despite lower salinity and lower temperature, CDOM optical properties in PSW in this study did not differ significantly from AW in 2013 and 2015, and similar variability ranges of CDOM absorption coefficient and spectral slope coefficient were reported by Pavlov at el. (2017) north of Svalbard. "

L. 549: What do the authors mean by "Statistical distribution"?

**Response:** Indeed, "statistical distribution" is not an appropriate expression in this sentence. The authors mean the average values and ranges of values listed in the Table 2. The sentence has been rewritten as follows:

"Average values of $a$CDOM(350) in 2014 in PSW (Table 2) were similar with Arctic Waters north of the Polar Front in Barents Sea described by Hancke et al. (2014) and slightly higher than observed in this study in 2013 (0.32±0.16 m$^{-1}$) and 2015 (0.26±0.09 m$^{-1}$)."

L. 553: How did the authors infer that the waters are from different origins based on their dataset?

**Response:** This discussion sentence was not based our data set but on data presented in the literature. The DOM composition in the western part of Fram Strait in EGC transporting the PW southward is compositionally different from AW in the eastern part of Fram Strait. According to results presented by Jorgensen et al., (2014) and Gonçalves-Araujo et al 2016, humic substances dominated the composition of DOM in EGC, and the fluorescence intensity of humic-like substances was 3 to 5 times higher in these waters than in the AW in eastern part of Fram Strait. As the values coefficient describing CDOM optical properties in waters north of Svalbard were similar to those in Nordic Sea, we could speculate that water masses classified by hydrographic properties as PW north of Svalbard could have different DOM composition. This is consistent with findings by Gonçalves-Araujo et al 2018 who found very low CDOM absorption in the Amundsen and Nansen basins.

To avoid speculations we had deleted this sentence.
" PW in the ESC in eastern part of Fram Strait was optically different from PW in EGC in western part of Fram Strait and thus likely of different origin"

L. 556: Gonçalves-Araujo et al 2016 has studied FDOM and not CDOM as mentioned in the text.

**Response:** Corrected accordingly, the reference has been deleted.
"According to Hancke et al. (2014) the CDOM pool in the Barents Sea was predominantly of marine origin, while several studies show terrestrial CDOM in the PW of EGC (Granskog et al., 2012, Pavlov et al., 2015; ) and $a$CDOM(350) reported for PW in the EGC was significantly higher, by factor 2, than values reported in this study around Svalbard."

L. 565-566: Have the authors tried to look for the a350 vs. temperature correlations?

**Response:** We have been analyzing the correlations the $a_{CDOM}350$ vs. temperature. The results showed very week, negative correlation between $a_{CDOM}(350)$ vs. temperature, especially in 2014 and 2015 data sets however due to low values of determination coefficient we decided not to present these results.

L. 571-572: where did the authors take this information from?? Was it from the literature (then it is missing a citation) or from their dataset (then I would like to know how the data supports such an affirmation)?

**Response:** This speculative sentence has been deleted from the revised manuscript.

L. 573: "S300-600 varied little between water masses in a given season..." Have you tested it for significance?

**Response:** According to the Reviewer's suggestion, we have checked the significance between water masses in a given season with use of T test. The results are presented in the Table r2. The only statistically significant difference is IW / DW vs. other water masses in 2015.

Table r2: Results of T test of S300-600 grouped by year and water masses. Table list results of t-test that measure significance in differences in mean value. The difference between variable averages in selected layer are significant if significance level p<0.05. NS - not significant, S – signify cant differences.

| Year | Variable | Layer | t-value | df | p | Significance |
|---|---|---|---|---|---|---|
| 2013 | S300-600 | AW vs.PSW | 0.524860 | 44 | 0.602315 | NS |
| | | AW vs.PSWw | 1.118054 | 74 | 0.267160 | NS |
| | | PSW vs. PSWw | 0.064696 | 34 | 0.948795 | NS |
| 2014 | S300-600 | AW vs.PSW | -0.405673 | 176 | 0.685476 | NS |
| | | AW vs.PSWw | 0.874175 | 200 | 0.383071 | NS |
| | | AW vs.AAW | 0.240337 | 176 | 0.810348 | NS |
| | | AW vs.IW/DW | 1.881482 | 183 | 0.061494 | NS/close |
| | | PSW vs. PSWw | 0.811732 | 30 | 0.423340 | NS |
| | | PSW vs.AAW | 0.713604 | 6 | 0.502273 | NS |
| | | PSW vs.IW/DW | 1.561149 | 13 | 0.142494 | NS |
| | | PSWw vs.AAW | -0.123383 | 30 | 0.902627 | NS |
| | | PSWw vs.IW/DW | 1.316258 | 37 | 0.196184 | NS |
| | | AAW vs.IW/DW | 1.058590 | 13 | 0.309061 | NS |
| 2015 | S300-600 | AW vs.PSW | 0.455974 | 160 | 0.649027 | NS |
| | | AW vs.PSWw | 1.928425 | 227 | 0.055050 | NS/close |
| | | AW vs.AAW | 2.012286 | 163 | 0.045837 | S |
| | | AW vs.IW/DW | -2.89410 | 173 | 0.004292 | S |
| | | PSW vs. PSWw | 0.193909 | 77 | 0.846757 | NS |
| | | PSW vs.AAW | 0.752780 | 13 | 0.464996 | NS |
| | | PSW vs.IW/DW | -1.49834 | 23 | 0.147646 | NS |
| | | PSWw vs.AAW | 0.900161 | 80 | 0.370736 | NS |
| | | PSWw vs.IW/DW | -3.14968 | 90 | 0.002219 | S |
| | | AAW vs.IW/DW | -2.86486 | 26 | 0.008150 | S |

L. 612: "...could indicate freshly produced CDOM." . At what depth was this found? Have you looked at the correlation between S300-600 and Chla? Is there any correlation for the high Chla samples?

**Response:** We have not observed any correlation between $S_{300-600}$ and chlorophyll $a$ concentration even for high Chla samples. The lower values of $S_{300-600}$ (<18 $\mu m^{-1}$) with higher absorption (>0.15 $m^{-1}$) were found at an average depth 30.81±33.26 m (min- max:: 0 - 129 m)

L. 611-614: This is an important discussion of the main results and could be discussed more in deep

**Response:** We appreciate Reviewer#2 comments, however, we have been discussing CDOM origin and its potential source in the Discussion, subsection 4.2, where in several paragraphs we have provided arguments that lead us to conclusion that CDOM is predominantly of autochthonous origin in the Nordic Seas.

L. 617-620: Not clear whether the authors have checked that or it is only an assumption.

**Response:** We have checked the location of points outside the model. The points were scattered and did not have one precise location, most of them were in places that could indicate the contribution from land: for example, points located at the mouth of the fjords or under the influence of low saline PW. Especially in 2015, all points outside the model limits were placed in cold and low saline water.

L. 623-625: Not clear what the authors wanted to state in that sentence.

**Response:** It is a statement taken from the cited reference. It leads to a discussion about SUVA$_{254}$, with very low values potentially associated with very low aromaticity of DOM in Nordic Seas.

L. 654: Authors may want to look at the results from Walker et al 2013 for comparison.

**Response:** We have added citation to Walker et al., 2013. In context of the first paragraphs of subsection 4.3, we have thought about that paper by Fichot et al, 2013, Scientific Reports, 3 : 1053 | DOI: 10.1038/srep01053, which presents use of remote sensing methods for tracing terrigenous organic matter in the Arctic Ocean.

"The riverine input can be monitored by optical methods with absorption, fluorescence or remote sensing measurements (Spencer et al., 2012; Walker et al., 2013; Fichot et al., 2013; Mann et al., 2016)."

Reference to [papers by Fichot et al., 2013 and Walker et al., 2013 have been added to reference list.

Fichot, C. G., K. Kaiser, S. B. Hooker, R. M. W. Amon, M. Babi, S. Bélanger, S. A. Walker, and R. Benner. Pan-Arctic distributions of continental runoff in the Arctic Ocean. , Scientific Reports, 3: 1053, DOI: 10.1038/srep01053, 2013.

Walker, S. A., R. M. W. Amon, and C. A. Stedmon (2013), Variations in high-latitude riverine fluorescent dissolved organic matter: A comparison of large Arctic rivers, Journal of Geophysical Research Biogeoscience, 118, 1689–1702, doi:10.1002/2013JG002320

L. 682-683: Why do the authors think the a350* vs. S300-600 behaves like this? Was the correlation also strong (and significant) for each year analyzed separately? It does not look like that (specially for 2013) when looking at Fig. 8.

**Response:** The non-linear, hyperbolic type of relationship between spectral slope and spectral values of specific (and also bulk) absorption coefficient are commonly reported in many papers (e.g. Fichot and Benner 2011, 2012, Fichot et al, 2013, Norman et al., 2011). This effect is similar to effect of mixing of two water bodies with contrasting CDOM optical

properties, explained theoretically by Stedmon and Markager (2003). One should note that distribution of values of specific absorption coefficient across of variety of aquatic environments is similar to CDOM absorption coefficient distribution pattern. The highest $a_{CDOM}^*(\square)$ values were associated in fresh water environments: swamps rivers, humic lakes, and the smallest were reported in oligotrophic subtropical oceanic gyres – the same global pattern applied to $a_{CDOM}(\square)$ (see Massicotte et al., 2017). Secondly, this kind of relationship is regulated by DOM chemical structures – more humic, more aromatic DOM is characterized by higher values of $a_{CDOM}^*(\square)$, compared to autochthonous, marine DOM that has lower saturation with aromatic rings and has more aliphatic structures. The only difference between ours and by Fichot and Benner results, is the choice of spectral range used for spectral slope calculations. However, our results were consistent with the $a_{CDOM}^*(375)$ vs $S_{300\text{-}650}$ relationship presented by Norman et al., 2011. It seems that in an environment influenced by terrigenous DOM a hyperbolic relationship works better for spectral slope calculated over 275-295 nm spectral range, while in oceanic environments it is better to use a longer spectral range for slope calculations. We have used whole data set to derive our relationship. Of course, the use of smaller data sets for specific years will produce different results with lower values of determination coefficient.

L. 695-696: significant differences for humic-like and PTN-like FDOM...With respect to what were these differences?

**Response:** This was misfortunate phrase. We referred to global pattern of distribution of humic-like and protein-like fluorescence intensities values presented in cited references. The sentence has been rewritten for clarity as follows.

"The pattern distribution of fluorescence intensities of main FDOM components with depth in the global oceans' biogeochemical provinces is significantly different for humic–like and protein–like FDOM (Stedmon and Nelson, 2015, Catalá et al., 2016)."

L. 699-701: How do the authors explain such behavior?

**Response:** It is beyond the scope of this study to explain the global distribution pattern of main fluorophores in the ocean. This has been presented by e.g. Jorgensen et al., 2011, Kowalczuk et al., 2013, and Stedmon and Nelson, 2015, Catalá et al., 2016 and Yamashita et al., 2017. This behavior is the complex result of photobleaching, in situ production and microbial processing and vertical mixing. All those processes have different time scales and intensity in a given biogeochemical ocean province and produce specific type of vertical distribution of fluorescence intensity of given fluorophore types. Reviewer#2 may find detailed information in provided reference examples. We have cited them because we have also presented examples of distribution of ICH1 and ICH3 with depth, comparing our results to global patterns.

L. 702-705: not clear if the authors are presenting their own results or results from other studies.

**Response:** Citations indicated that we have been referring to published results from other studies. We have rewritten this sentence for clarity.

**"**The global pattern of fluorescence intensity of protein–like FDOM distribution across the oceanic biogeochemical provinces and with depths was opposite compared to humic–like

FDOM. Protein–like FDOM fluorescence intensity usually increased toward the open ocean and the highest intensity was observed in the surface waters, rapidly decreasing with depth, reaching constant low level below the epipelagic layer (Jørgensen et al., 2011; Kowalczuk et al., 2013; Catalá et al., 2016)."

L. 722-723: It is not completely clear based on the results presented here. The authors may want to rephrase that sentence. The authors state that the PTN-like FDOM is the dominant fraction at high salinity. However, studies have shown that the humic-like (visible fluorescence) can be the dominant signal at sal>34 in the Arctic Ocean.

**Response:** We partially agree with the Reviewer's suggestion. The humic-like FDOM may be significant in some Arctic Ocean Basins, but not in the Nordic Seas, where influenced by warm Atlantic Water as we shown in our results. There is hardly any sources of terrigenous origin to be found in these Atlantic dominated waters. We have rewritten this sentence for clarity,
"
"The distribution of fluorescence intensity of main FDOM components in the Nordic Seas, dominated by warm water of Atlantic origin followed the general trends observed globally."

L. 732-734: This sentence does not help the discussion and could be easily removed.

**Response:** According to the reviewer's suggestion, the sentence:
"The PW flowing through the Canadian Arctic Archipelago was enriched with humic–like component compared to Labrador Sea (Guéguen, et al., 2014)." has been deleted from the discussion section.

L. 742: The paper from Gonçalves-Araujo et al 2016 does not show CDOM results.

**Response:** Corrected accordingly.

"This was also in a good agreement with CDOM distribution in the Fram Strait (Granskog et al., 2012; Pavlov et al.; 2015, ) and FDOM humic–like fraction 743 (Ex/Em = 280/450 nm) distribution presented by (Granskog et al., 2015)."

L. 756: "...and same dominant factor controlling these parameters in time and space." How do your results support that statement?

**Response:** It is broad and well established consensus in ocean optics community that following optical parameters: chlorophyll *a* fluorescence intensity, $I_{FChla}$ and total non–water absorption coefficient at 676 nm, $a$tot–w(676) are regarded as optical proxies for phytoplankton biomass. In the main body of this manuscript and supplementary materials we have shown that both: $I_{FChla}$ and $a$tot–w(676) were strongly and significantly correlated with $I_{CH3}$. Therefore a tight coupling between those parameters and overlapping distribution with depth, justified our statement and is fully supported by our data set..

L. 762-763: Could the authors identify what FDOM fractions are produced through each of the mentioned processes (e.g. phytoplkt extracellular release, phytoplkt degradation or lysis)??

**Response:** Based on information provided in the literature we just name the processed known to lead to release of DOM by phytoplankton into adjacent waters, without specifying which process produce specific FDOM fraction. It would be a subject of further studies. For clarity we have deleted this sentence and we have removed citations from reference list.

""